# 3D quantification of nanolites using X-ray ptychography reveals syn-eruptive nanocrystallisation impacts magma rheology

Emily C. Bamber [1,2] ✉, Fabio Arzilli[3], Silvia Cipiccia [4,5], Darren J. Batey[5], Giuseppe La Spina [6], Margherita Polacci [2], Ali Gholinia [7], Heath Bagshaw[8], Danilo Di Genova [1], Richard Brooker [9], Daniele Giordano[10], Pedro Valdivia [1,11] & Mike R. Burton [2]

Nanoscale crystals are becoming increasingly recognised in the products of volcanic eruptions, spanning a range of magma compositions. The crystallisation of nanolites impacts magma rheology, ascent dynamics, and eruptive style. Their impact can be enhanced due to their capacity to aggregate and develop neighbouring chemically differentiated boundary layers. However, their 3D interaction, spatial distribution, and morphology is not currently understood. Here we present a cutting-edge, 3D nanometre-scale visualisation and quantification of nanolites in scoriae of the Las Sierras-Masaya basaltic Plinian eruptions, acquired using X-ray ptychography. We find that Ti-magnetite nanolites aggregate, forming elongate, irregular structures in 3D. Their crystallisation extracts Fe and Ti from the melt, resulting in differentiated boundary layers with higher viscosity. Syn-eruptive crystallisation of nanolites and their interaction is estimated to have increased magma viscosity by 2–3 orders of magnitude, therefore, they likely had a strong control on magma rheology, increasing the potential of magma fragmentation.

The potential of a highly explosive Plinian eruption at a volcanic system presents a significant hazard and requires assessment of risk mitigation. Plinian eruptions eject >1 km³ of tephra into the atmosphere, producing eruption columns which exceed 10 km in height[1], with considerable impacts on society and the environment. Plinian eruptions are typically associated with silica-rich magmas[1], as their high viscosity can result in brittle fragmentation within volcanic conduits, which occurs due to the accumulation of stress at high applied strain rates or the overpressure within entrained bubbles[1–3]. However, sub-Plinian and Plinian eruptions at basaltic volcanic systems do occur, such as the recent 2017–2018 eruption of Ambae (Aoba), Vanuatu, which produced a maximum column height of 20 km and led to the

¹Institute of Science, Technology and Sustainability for Ceramics (ISSMC), National Research Council (CNR), Via Granarolo 64, 48018 Faenza, Italy. ²Department of Earth and Environmental Sciences, The University of Manchester, Oxford Road, Manchester, UK. ³School of Science and Technology, Geology Division, University of Camerino, Camerino, Italy. ⁴Department of Medical Physics and Biomedical Engineering, University College London, London, UK. ⁵Diamond Light Source, Harwell Science and Innovation Campus, Fermi Avenue, Didcot, UK. ⁶Istituto Nazionale di Geofisica e Vulcanologia - Osservatorio Etneo, Sezione di Catania, Piazza Roma, 2, Catania, Italy. ⁷Department of Materials, The University of Manchester, Oxford Road, Manchester, UK. ⁸School of Engineering, The University of Liverpool, Liverpool, UK. ⁹School of Earth Sciences, University of Bristol, Bristol, UK. ¹⁰Department of Earth Sciences, University of Turin, Via Valperga Caluso, 35, Turin, Italy. ¹¹Bayerisches Geoinstitut, University of Bayreuth, Universitätsstraße 30, Bayreuth, Germany.
✉e-mail: emilycharlotte.bamber@issmc.cnr.it

evacuation of 11,000 people[4,5]. Las Sierras-Masaya volcanic system (Nicaragua), located approximately 25 km from the capital city Managua, has produced several basaltic Plinian eruptions in the past 60 ka[6] and, therefore, may have conditions which favour recurrent highly explosive activity[7,8].

The low viscosity of basaltic magma often prevents the accumulation of sufficient overpressure to result in fragmentation, unless subject to exceptionally high strain rates during ascent[9]. However, processes such as fracturing[10], the inertial expansion of bubbly magma[11] and rapid syn-eruptive crystallisation[12,13] can promote the fragmentation of basaltic magma.

Syn-eruptive crystallisation increases magma viscosity through the physical presence of crystals, crystal-crystal interactions[14] and the facilitation of heterogeneous bubble nucleation[15]. The fast ascent rates estimated for basaltic Plinian eruptions may induce large undercooling ($\Delta T$) during ascent, favouring rapid, syn-eruptive microlite crystallisation and bubble nucleation[7,8,12,16–18]. The erupted products often show heterogeneous and highly crystalline microtextures[7,8,17,19]. Crystallinity varies between 43 and 99 vol.% in clasts of the Etna 122 BC eruption (Italy)[16], although the syn-eruptive crystal content may be lower. Furthermore, inspection of more crystal-poor regions of some clasts also shows a mottled texture[16,17,20], indicating that even apparently glassy areas may show evidence of incipient crystallisation at high magnification.

Nano-sized crystals, or 'nanolites', are becoming increasingly recognised[21] in both experimental products and natural volcanic samples, encompassing a range of magma compositions and explosivity[13,20,22–33]. Nanolites 20-50 nm in size have been observed in clasts of the Etna 122 BC basaltic Plinian eruption[13]. Nanolites can considerably increase magma viscosity through the rheological impact of the nanoparticle suspension, by facilitating heterogeneous bubble nucleation or inducing chemical and structural changes in the residual melt, for example, by increasing melt polymerisation and/or creating a heterogeneous distribution of chemical components[13,21,27–29,32,34–40]. They may, therefore, exert an important control on rheology and ultimately, explosivity. Nanolite crystallisation may also hinder laboratory-based measurements of melt viscosity, resulting in an overestimation of the crystal-free melt viscosity in magma compositions prone to nanocrystallisation[23,35,38,40].

The rheological impact of nanolites is thought to be disproportionate with respect to their size and often low particle content in magmas[13,41]. The high number density ($N_a$) of nanoscale crystals can influence liquid flow between particles, whilst the local development of differentiated, high viscosity residual melt surrounding nanolites can induce rheological changes even at a low particle fraction[35,36,38,40]. Due to their high surface area, nanolites are also susceptible to particle aggregation through van der Waals interactions, a process which considerably increases the viscosity of nano-bearing suspensions[13,41]. As aggregation will also incorporate the melt between nanolites, their effective volume increases, thereby increasing the viscosity of nanolite-bearing magma[13]. However, so far, the aggregation of nanolites has been invoked, but not observed.

Despite the potential influence of particle interactions on the rheology of nanolite-bearing suspensions, the interaction between nanolites has yet to be examined using a three-dimensional (3D) approach. It is important to investigate the spatial distribution, interaction, and aggregation of nanolites in 3D, in to order to evaluate the impact of their crystallisation on conduit dynamics and eruptive style. However, the technical challenge of resolving nano-sized crystals in multiphase, geological materials, where crystals and surrounding glass have comparable contrast, renders it difficult to resolve and visualise nanolites in 3D using more conventional tomographic techniques[42].

X-ray ptychography is a lensless, phase imaging microscopy technique at the nanoscale, also described as scanning coherent diffraction imaging (CDI) (Fig. 1). A ptychographic acquisition consists of scanning a sample across a coherent beam whilst recording the diffraction patterns. These are inverted to form the image of the sample using an algorithm instead of a lens. In standard CDI, the field of view is limited by the lateral coherence length of the beam. Instead by scanning, ptychography overcomes this limitation, allowing the imaging of extended objects.

Ptychography was first conceived to solve the lens aberrations in electron microscopy[43,44] and then, in 2007, successfully applied in the X-ray regime[45]. Since then, X-ray ptychography has achieved a resolution that exceeds standard X-ray optics[46] and is amongst the most used techniques for nanoscale imaging at synchrotron facilities. Thanks to its quantitative nature, robustness, and high sensitivity to changes in electron density ($\rho_e$), X-ray ptychography has enabled scientific breakthroughs across a wide range of disciplines from magnetic material studies[47] to micro-electronics[46,48].

We present an innovative 3D reconstruction and visualisation of nanolites in scoriae of basaltic Plinian eruptions, acquired using X-ray ptychography. By reconstructing nanolites in 3D, we can visualise and quantify their spatial distribution, interaction, and relation with the residual melt, evaluating their impact on magma rheology during ascent and eruption. We find that nanoparticles are prone to aggregation in basaltic magmas, increasing their effective volume. Furthermore, nanolite crystallisation produces chemical changes in the surrounding melt, which leads to the development of localised boundary layers of higher viscosity melt surrounding nanolites. The crystallisation and subsequent aggregation of nanolites markedly increase their overall contribution to magma viscosity, inducing rheological changes during ascent that may promote fragmentation and a highly explosive eruption. Our results provide insight into the nanoscale structure of volcanic products, and also contribute to our understanding of the driving mechanisms of basaltic Plinian eruptions, amongst the most hazardous styles of volcanic activity.

## Results and Discussion
### 2D observations of nanolites
A range of imaging techniques were used to examine textural features in scoriae of two basaltic Plinian eruptions of Las Sierras-Masaya volcanic system; the Fontana Lapilli (FL) and Masaya Triple Layer (MTL) eruptions[7,8] (Methods). High resolution back-scattered electron (BSE) images obtained using scanning electron microscopy (SEM) show nanoscale crystals in FL scoriae. Due to their size (<1 μm), these crystals can be classified as nanolites according to the scheme of Mujin et al.[25]. Nanolites are found in areas where there is interaction between melts of two different compositions (Fig. 2a, b), gradually increasing in size towards the contact (Fig. 2c, d). Nanolites also form structures with concave inner surfaces and more irregular outer contacts (Fig. 2e, f), as observed in FL scoriae, which may represent shells of oxides that nucleate on the surface of bubbles during vesiculation at the bubble-melt interface[49]. The occurrence of multiple shells may reflect a sequence of detachment from bubbles and subsequent infilling by melt, crystallising multiple generations of oxides and microlites[49]. Their size increases towards the contact with the glass matrix. Crystal-poor glass also shows a mottled appearance (Fig. 2b, d), possibly due to the presence of nanolites[16,17,20], but where the size of the individual crystals is below the resolution of the SEM.

Analysis of the BSE images provides an estimated nanolite crystal fraction ($\Phi$) of 0.27 ± 0.02. Simulations using Rhyolite-MELTS (v.1.2)[50] and the average FL bulk composition find that the measured nanolite crystal fraction (0.27 ± 0.02) can be reached at a temperature of 1040 °C and a pressure of 10 MPa, with fO$_2$ equivalent to the QFM buffer. These conditions are within the estimated range for the FL eruption[8]. The simulated crystal content consists of both clinopyroxene (0.21) and titanomagnetite (0.06), suggesting that the measured FL nanolite crystal fraction of 0.27 from the 2D BSE images may consist of both phases. The maximum titanomagnetite crystal

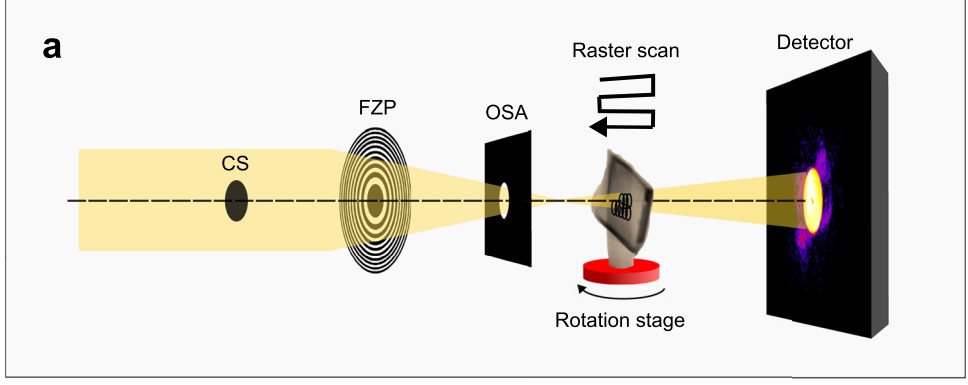

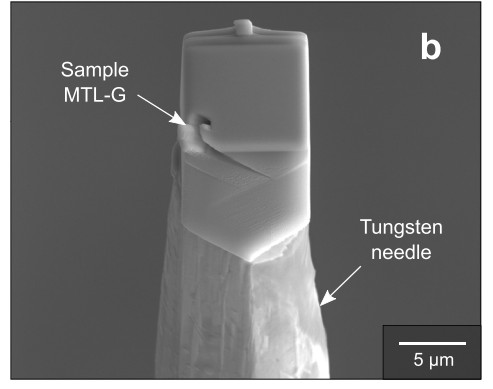

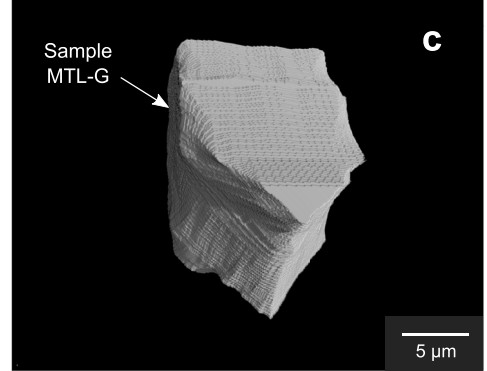

**Fig. 1 | Experimental setup for acquisition using X-ray ptychography. a** A diagram illustrating the experimental setup of the ptychography end-station at I13-1 (CS = central stop, FZP = Fresnel zone plate, OSA = order sorting aperture). Also shown is **b** the sample MTL-G, a 10 μm sized cube extracted from silicate glass within a clast of the MTL eruption using FIB, mounted on a tungsten needle. **c** 3D rendering of silicate glass sample MTL-G, produced from the data acquired using X-ray ptychography. The volume rendering was produced using VGStudio.

fraction estimated to crystallise from the melt is 0.15, as even for the Fe-rich FL melt composition, complete extraction of Fe from the melt during nanolite crystallisation may only crystallise a small vol.% of titanomagnetite[38]. The evolved melt composition (Supplementary Table 1) shows a depletion also in MgO and CaO.

By using high angle annular dark field (HAADF) scanning transmission electron microscopy (STEM) imaging, we can resolve individual nanolite crystals (Fig. 3) with higher resolution. Images of MTL samples acquired using STEM show spherical-subspherical nanoscale particles 20-30 nm in diameter (Fig. 3d), which can be described as ultrananolites[25]. These spherical-subspherical particles also form clusters with more elongate, irregular shapes (Fig. 3e), possibly resulting from the aggregation of individual nanolites. Both textures are comparable to the spherical and elongate structures observed in transmission electron microscopy (TEM) and STEM images of pumice of the Havre 2012 eruption, where irregular, elongate structures are interpreted as nanolite aggregates[30]. Instead, MTL samples that consist of only silicate glass (Fig. 3f) do not show these features.

Elemental mapping using energy-dispersive X-ray spectroscopy (EDS) and STEM shows that the spherical-subspherical clusters observed in the HAADF-STEM images have a higher concentration of Fe (Fig. 3g, h) and Ti (Fig. 3j, k) than the surrounding silicate glass. Fe and Ti X-ray intensity maps show the same clustering of spherical-subspherical nanoscale particles as observed in the HAADF-STEM images (Fig. 3g, j) and also elongate, irregular Fe-rich and Ti-rich structures where nanolite aggregates occur (Fig. 3h, k). Instead, X-ray intensity maps of the silicate glass sample show no relative increase in the intensity of Fe and Ti (Fig. 3i, l). Similarly, Si X-ray intensity maps of the three samples show no increase in the intensity of Si relative to that expected for homogeneous silicate glass, as the intensity of Si is comparable for both the nanolite-bearing and nanolite-free samples (Fig. 3m-o).

FL and MTL scoriae contain phenocrysts (>40 μm in size) and microlites (<40 μm) of plagioclase, clinopyroxene, olivine and titanomagnetite[6–8,19]. Considering the micron-scale mineral phase assemblage, X-ray elemental intensity maps and HAADF-STEM images, the observed Fe and Ti-rich, spherical-subspherical nanolites most likely represent titanomagnetite.

## 3D reconstruction of nanolites using X-ray ptychography

To investigate the morphology and distribution of nanolites in 3D, three samples of the MTL eruption were examined using the ptychography end station of the I13-1 beamline[51] at Diamond Light Source, Harwell Science & Innovation Campus (United Kingdom) (Methods). Sample MTL-G does not contain nanoscale particles within the sample volume and was used as a reference sample of homogeneous silicate glass (of basaltic composition) to compare with the nanolite-bearing samples MTL-A and MTL-B.

The 3D images of samples MTL-A and MTL-B show spherical-subspherical nanolites with a higher electron density ($\rho_e$) than the surrounding silicate glass. While the $\rho_e$ depends on both the density ($\rho$) and composition (and therefore, atomic number) of the crystals and glass[42], since this is a solid phase, we expect the measured $\rho_e$ to be predominantly affected by the difference in composition. The HAADF-STEM images, combined with the X-ray intensity maps, illustrate that the nanolites are Fe- and Ti-rich (Fig. 3g-h, j-k). In the orthoslices, spherical-subspherical nanolites can be resolved as they have a higher $\rho_e$ than the silicate glass (Fig. 4). The spherical-subspherical shapes of the nanolites are consistent with previous studies that examine nanolites in volcanic samples using high-resolution TEM and STEM imaging[13,25,30] and also the STEM images of the MTL scoriae (Fig. 3d, e).

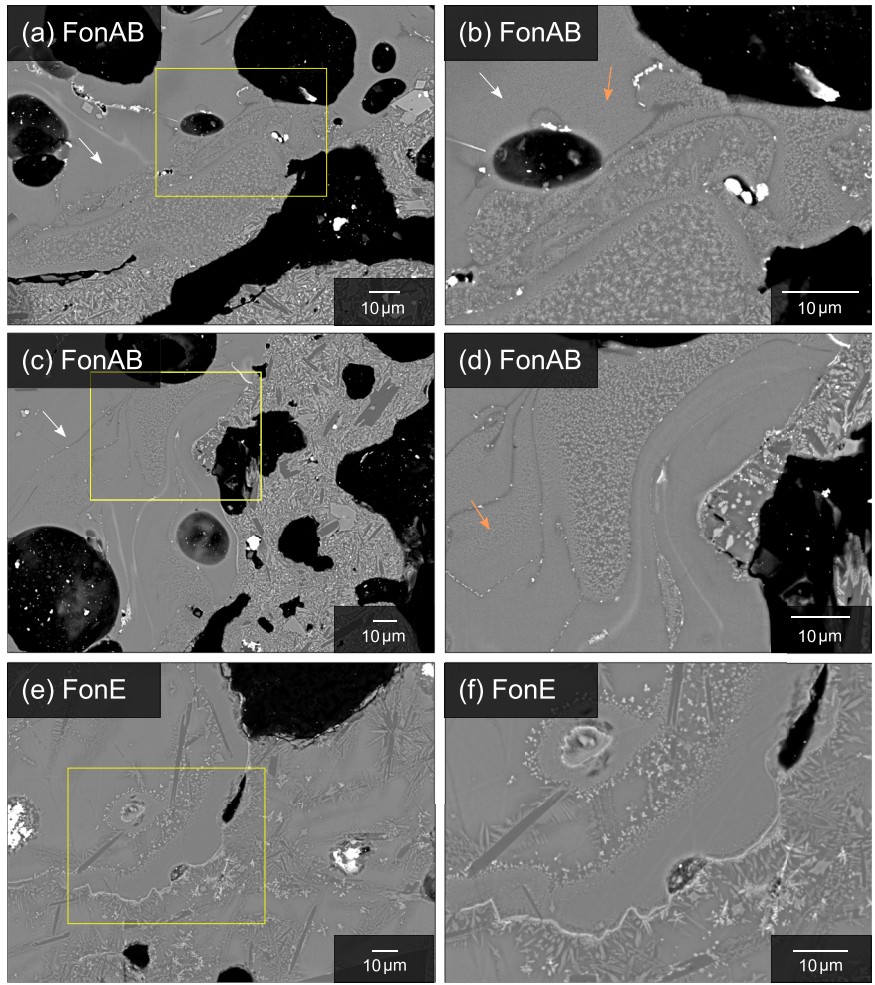

**Fig. 2 | Back-scattered electron (BSE) images showing the distribution of nanolites in scoriae of the FL eruption.** Nanolites occur (**a**, **c**) at the boundaries between two different melts as spherical-subspherical crystals. The white arrows indicate the more evolved silicate melt composition. At higher magnification, (**b**, **d**) the crystal-poor glass shows a mottled texture (highlighted by the orange arrows), which may indicate the presence of nanolites, but of a size which is below the resolution of the technique. Nanolites are also observed along the (**e**, **f**) boundaries of concave structures, which may represent shells consisting of oxide crystals that nucleate on the surface of bubbles, where the bubble is then lost by detachment and not preserved[49]. The sequence of concave structures may represent several generations of oxide nucleation on bubbles, forming multiple shells.

For the nanolite-bearing samples MTL-A and MTL-B, the ptychographic images show that $\rho$, and, therefore, composition, varies within the glass. Orthoslices of the ptychographic reconstructions show a halo surrounding nanolites (Fig. 4), corresponding to an area with a lower $\rho_e$ than that of the nanolite-free glass in sample MTL-G (Figs. 4, 5; Supplementary Fig. 1). The lower $\rho_e$ suggests that these areas have a different composition and lower $\rho$ than expected for a homogeneous silicate glass. This variation in $\rho_e$ may reflect the formation of differentiated compositional boundary layers that can develop at the melt-crystal interface during crystal growth[33,52–57]. As these layers result from the crystallisation of Fe-Ti oxide, the layers will be enriched in incompatible elements such as Si and Al and depleted in Fe and Ti[21,33,40,57]. This agrees with the lower $\rho_e$, and thus, $\rho$ of the boundary layer, as the more evolved, residual melt enriched in $SiO_2$ and $Al_2O_3$ will have a lower density than the bulk composition[58].

The commercial software Avizo (v.2019.1; Thermo Fisher Scientific, USA) was used to segment nanolites from the silicate glass and reconstruct their 3D morphology and spatial distribution. In 3D, nanolites form aggregated structures, where several particles are often joined together to form elongate structures with a more irregular morphology (Fig. 6a, b). This morphology is comparable to the 2D observations of nanoscale aggregates in clasts sampled from other volcanic eruptions, such as the Havre 2012 pink pumice[30] and the Etna 122 BC eruption[13]. In the MTL

samples, nanolites are approximately 20–300 nm in diameter where they are isolated. Instead, aggregates range from 400 nm to 6 μm in size (Figs. 4, 6; Supplementary Tables 2-3). Nanoscale structures observed in the orthoslices of the 3D reconstructed volumes are of comparable size and shape to those observed in HAADF-STEM images (Fig. 3). By performing a second segmentation, which accounts also for the more differentiated boundary layers surrounding nanolites (Fig. 6c, d), the initial volume increases by 60%.

A comparative analysis of 2D and 3D characterisation methods (Supplementary Table 4) indicates that estimates of the nanolite volume derived from 2D data may be underestimated by one to two orders of magnitude. This discrepancy arises as a 2D cross-section may not fully capture the nanolite morphology in 3D, resulting in a greater underestimation of the volume when nanolites exhibit complex, dendritic morphologies formed through aggregation-driven growth. Instead, stereological corrections of the volume using 2D images may provide more accurate volume estimates for smaller, isolated nanolites with spherical-subspherical shapes and low aspect ratios.

**Identification of nanolites**
The variation in the grayscale value in the orthoslices is a measure of the $\rho_e$ of the sample. The $\rho_e$ depends on both the material $\rho$ and composition, and thus, the atomic number. Therefore, this variation in

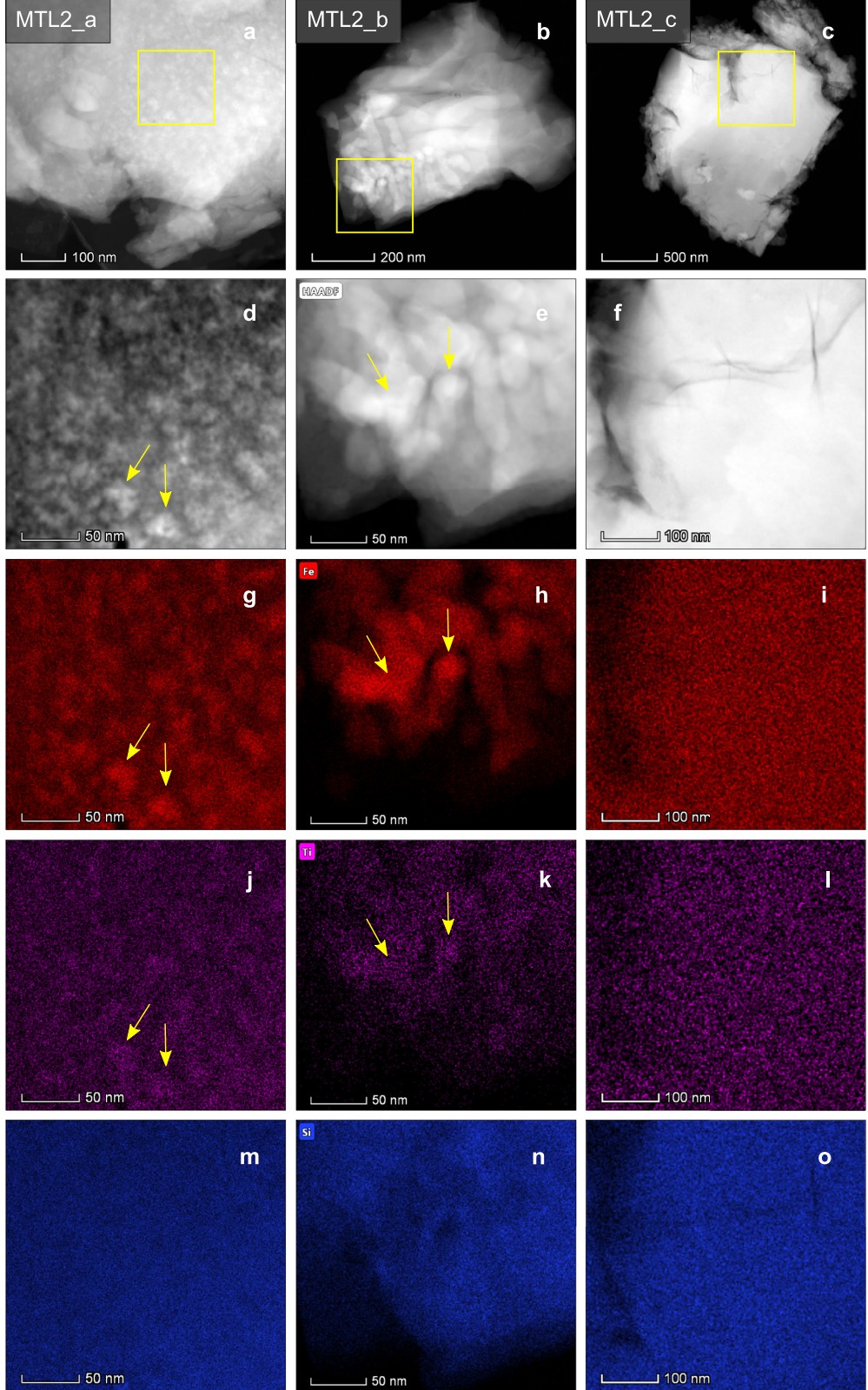

**Fig. 3 | HAADF-STEM images showing the distribution and chemical composition of nanolites in samples of the MTL eruption. a–c** HAADF-STEM images of three samples of MTL scoriae, showing the distribution of nanolites and silicate glass. At higher magnification, nanolites (indicated by the yellow arrows) can be observed in the HAADF-STEM images which form either **d** spherical-subspherical isolated nanolites or **e** aggregates which are more elongate and irregular in shape. Instead, **f** shows only silicate glass, where no nanolites are observed. Also shown are X-ray intensity maps, where a similar clustering of **g**, **h** Fe and **j**, **k** Ti is observed in areas where nanolites occur. Instead, the **i**, **l** silicate glass sample shows no relative increase in Fe and Ti. All samples show a similar distribution of Si from the X-ray intensity maps (**m–o**), which may result from the surrounding silicate glass.

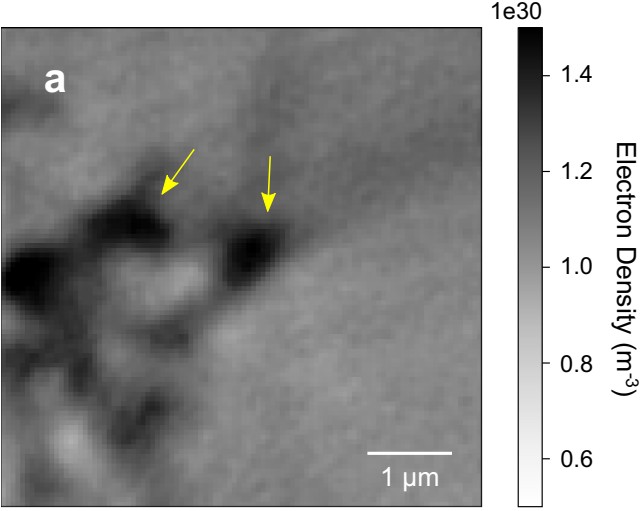

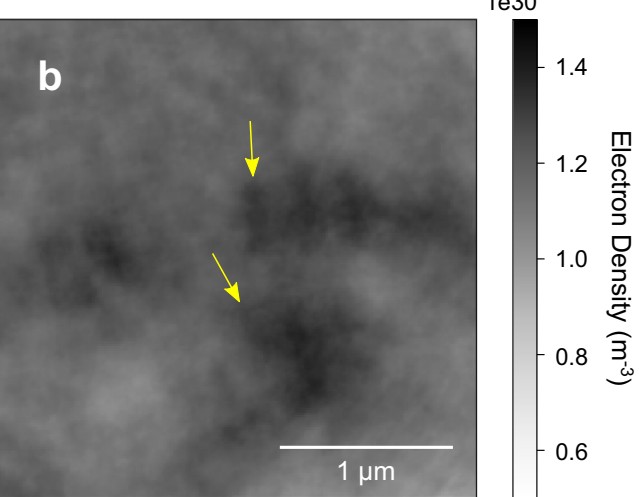

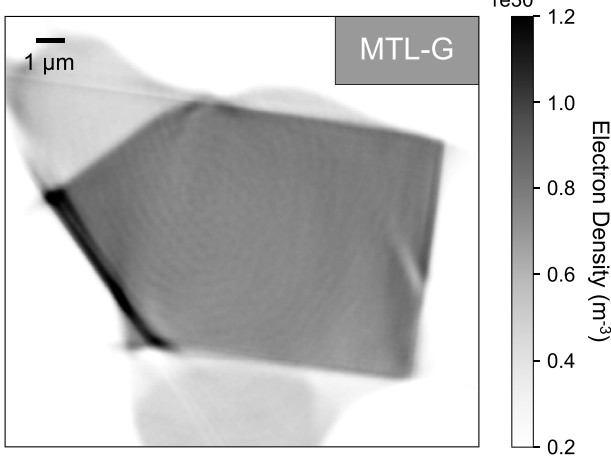

**Fig. 5 | Orthoslice of sample MTL-G acquired using X-ray ptychography.** The electron density is consistent with a nanolite-free, homogeneous silicate glass of basaltic composition ($\rho$ - 2700 kg m$^{-3}$).

**Fig. 4 | Orthoslices obtained from the 3D reconstructions acquired using X-ray ptychography.** Shown are samples **a** MTL-A and **b** MTL-B, where spherical-subspherical nanolites can be identified (indicated by the yellow arrows) due to their higher electron density when compared to the surrounding silicate glass. Nanolites form spherical-subspherical crystals where isolated, in addition to more elongate, aggregated structures.

$\rho_e$ can be used to identify the nanolite aggregates. We can relate the phase shift due to the X-ray beam passing through the material compared to the air, to the $\rho_e$ of the unknown phase. Equations 1 and 2[59] was used to calculate the $\rho_e$ (m$^{-3}$) of the nanoscale phase using the phase shift:

$$\rho_e = \frac{2\pi}{\lambda^2 r_e} \delta \qquad (1)$$

$$\delta = \frac{\lambda \phi}{2\pi d} \qquad (2)$$

where $\delta$ is the refractive index decrement, $r_e$ the classical electron radius, $\lambda$ the wavelength of the X-ray, $\phi$ the measured phase shift, and $d$ the reconstructed voxel size.

To identify the nanolite phase and to quantify the glass composition, the $\rho_e$ was then used to calculate the atomic density ($\rho_a$) (m$^{-3}$), for comparison with a reference material of known composition using

Eq. 3:

$$\rho_a = \frac{\rho_e}{Z} \qquad (3)$$

where $Z$ is the sum of the atomic number of each element of the molecule under consideration, multiplied by the respective subscript.

To calculate the reference $\rho_a$ for the basaltic glass and titanomagnetite composition, as expected for samples of the MTL eruption (Methods), we used the measured glass and titanomagnetite compositions acquired using electron probe microanalysis (EPMA) for the MTL eruption[7] and the average atomic number ($Z$) calculated by Probe for EPMA software (v.12.9.5) for the glass and titanomagnetite respectively[60]. Figure 4 shows orthoslices of the 3D reconstructions obtained using X-ray ptychography and the range in $\rho_e$ for both nanolites and glass.

For the nanolites, the estimate of $\rho_a$ shows good agreement with the reference $\rho_a$ calculated for magnetite, using a $\rho$ of 5200 kg m$^{-3}$[61] (Table 1). Combined with the results of the X-ray intensity maps (Fig. 3) and expected mineral phase assemblage for scoriae of the MTL eruption, the nanolite phase is most likely titanomagnetite. For sample MTL-G which consists only of homogeneous silicate glass, the calculated glass $\rho_a$ shows good agreement with the reference $\rho_a$ estimated for a glass with a $\rho$ of 2700 kg m$^{-3}$, comparable with the estimated $\rho$ for the MTL magma composition[8]. In contrast, overall, the glass in samples MTL-A and MTL-B does not have a similar $\rho_a$ to the homogeneous glass sample MTL-G, showing better agreement with a glass of higher $\rho$ ($\geq$3000 kg m$^{-3}$).

The discrepancy in $\rho_a$ calculated for the silicate glass matrix when comparing nanolite-free and nanolite-bearing samples further suggests that the variation in $\rho_e$ observed in the reconstructed 3D volumes is controlled by a variation in glass composition and, therefore, $\rho$. Interstitial glass between nanolites in samples MTL-A and MTL-B is of lower $\rho$ and a different composition compared to nanolite-free sample MTL-G. This may be due to a depletion in compatible elements such as Fe and Ti and an enrichment in Si and Al following nanolite crystallisation[21,40]. Similarly, a study on glass ceramics finds variations in $\rho_e$ following the crystallisation of Fe-Mn-O nanoparticles, where the extraction of Fe and Mn from the silicate melt produced a 'core' ($\rho$ = 4900 kg m$^{-3}$) enriched in Fe and Mn, with a higher $\rho_e$ than the Si-rich 'shell' ($\rho$ = 2200 kg m$^{-3}$)[57].

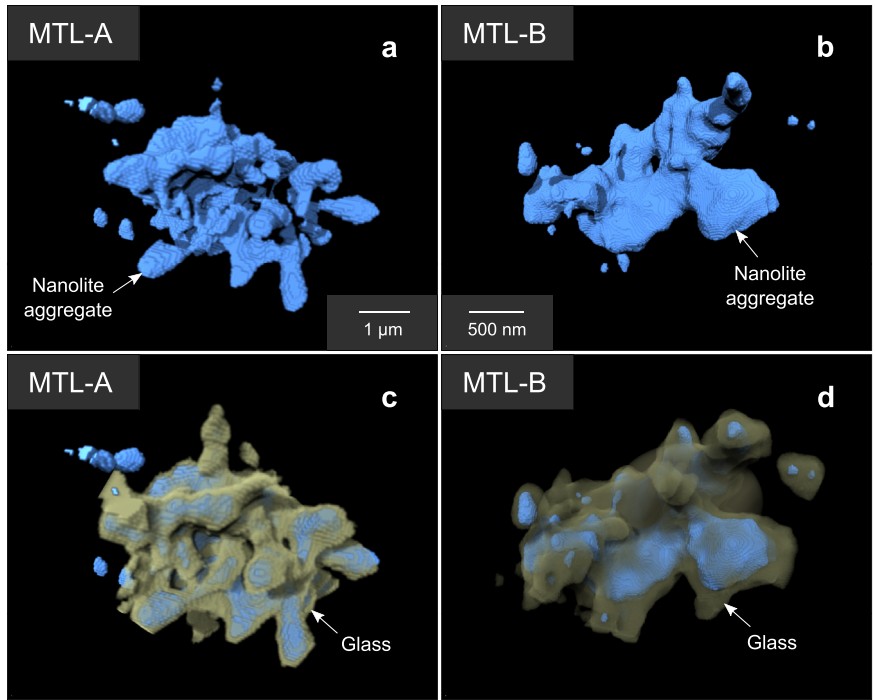

**Fig. 6 | Volume renderings of the nanolite aggregates observed in samples MTL-A and MTL-B. a** and **b** show volume renderings where only the nanolites were segmented, which occur as subspherical, isolated crystals as well as more irregular shaped aggregates. **c** and **d** show volume renderings where also the evolved, Fe-depleted glass around and between nanolites is segmented (highlighted in yellow). The segmentation of the nanolites, matrix glass and evolved boundary layer was performed based on visual inspection of the images and the estimates of $\rho_e$. All volume renderings were produced using VGStudio. In **c**, the rendering has a cut-away section to allow visualisation of the nanolites. The complete rendering is presented in the Supplementary Information (Supplementary Fig. 4).

### Table 1 | Results of the $\rho_e$ and $\rho_a$ calculation for the nanolite phase and glass

| Sample | MTL-A | MTL-B | MTL-G | Reference | $\rho$ (kg m$^{-3}$) |
|---|---|---|---|---|---|
| | *Nanolite* | | | *Nanolite* | |
| $n$ | 5 | 10 | | | |
| $\rho_e$ (m$^{-3}$) | $1.48 \times 10^{30}$ | $1.4 \times 10^{30}$ | | | |
| $\sigma$ | $1.5 \times 10^{28}$ | $1.48 \times 10^{28}$ | | | |
| $\rho_a$ (m$^{-3}$) | $1.45 \times 10^{29}$ | $1.38 \times 10^{29}$ | | $1.54 \times 10^{29}$ | 5200 |
| $\sigma$ | $1.48 \times 10^{27}$ | $1.46 \times 10^{27}$ | | | |
| | *Glass* | | | *Glass* | |
| $n$ | 10 | 10 | 10 | | |
| $\rho_e$ (m$^{-3}$) | $9.85 \times 10^{29}$ | $9.62 \times 10^{29}$ | $7.92 \times 10^{29}$ | | |
| $\sigma$ | $1.28 \times 10^{28}$ | $1.48 \times 10^{28}$ | $5.31 \times 10^{27}$ | | |
| $\rho_a$ (m$^{-3}$) | $1.53 \times 10^{29}$ | $1.49 \times 10^{29}$ | $1.23 \times 10^{29}$ | $1.26 \times 10^{29}$ | 2700 |
| | | | | $1.31 \times 10^{29}$ | 2800 |
| | | | | $1.35 \times 10^{29}$ | 2900 |
| | | | | $1.4 \times 10^{29}$ | 3000 |
| $\sigma$ | $1.98 \times 10^{27}$ | $2.3 \times 10^{27}$ | $8.22 \times 10^{26}$ | | |

Both the electron density ($\rho_e$) and atomic density ($\rho_a$) are presented as an average and standard deviation (1$\sigma$) for each 3D volume, calculated using a number of orthoslices ($n$). For comparison, the $\rho_a$ of a reference magnetite and basaltic glass composition from the MTL eruption are also presented, calculated using the estimated atomic number and density ($\rho$).

### Changes in melt composition due to nanolite crystallisation

Crystallisation of titanomagnetite nanolites will extract Fe from the melt structure, leading to structural modification of the melt due to the effect of Fe content and its valence state on the degree of melt polymerisation[26,62–64]. The depletion of iron and its effect on melt oxidation state, namely the change in the relative proportion of network-modifying $Fe^{2+}$ cations relative to network-forming $Fe^{3+}$ cations, and the subsequent increase in silica content, has been shown to increase melt viscosity[23,35,37,62–65]. Crystallisation of pyroxene

nanolites can have a similar effect, through the preferential extraction of network-modifying cations $Fe^{2+}$, Mg, and Ca[65].

The potential for nanolite crystallisation is controlled by melt composition, particularly its $TiO_2$ and $FeO_{tot}$ content[21,37]. The experiments of Scarani et al.[21] demonstrate that the Etna trachybasaltic composition ($FeO_{tot} = 10.05$ wt.%; $TiO_2 = 1.67$ wt.%) is more prone to nanolite crystallisation than the Stromboli basaltic magma composition ($FeO_{tot} = 7.58$ wt.%; $TiO_2 = 0.92$ wt.%), attributed to its higher transition metal oxide content. As magmas erupted at Las Sierras-

Masaya volcanic system are tholeiitic (MTL $FeO_{tot} = 13.67$ wt.%; $TiO_2 = 1.44$ wt.%)[7,8], they have an enrichment in Fe and Ti relative to calc-alkaline melt compositions, and consequently, may be more prone to nanolite crystallisation. The incipient stage of nanolite crystallisation involves clustering of Fe and Ti, gradually forming a nanoscale heterogeneous structure that eventually crystallises Fe and Ti-bearing nanolites[40]. This can also result in an enrichment in Al surrounding nanolites, and Si in the matrix[21,35,40]. Our HAADF-STEM images of the MTL samples show a similar clustering of Fe and Ti, forming subspherical particles that are 20-30 nm in diameter.

Studies on both volcanic samples and glass ceramics show that compositional boundary layers form surrounding micron-scale and nanoscale crystals, which develop during their growth[29,33,40,52-54,56,57,66], indicating that nanolites may influence the structure and chemistry of the surrounding matrix glass during crystallisation[13,21,29,36,38,40,67]. As the boundary layers are depleted in cations such as Fe, Mg, and Ti, yet enriched in the network-forming cations Si and Al, these layers may produce a local increase in viscosity relative to the bulk composition, due to the relatively homogeneous matrix formed from a stable network of Si and O bonds[21,29,36,38,40,65]. These differentiated layers can increase the effective volume of the nanolite, as their rheological behaviour will be closer to that of an elastic response[29,38]. In our samples, the estimated thickness of the differentiated glass layer surrounding nanolites ranges between 10 and 350 nm. Comparing the segmentation of the nanolite aggregates both with and without the incorporation of the boundary layer (Fig. 6), the initial volume of the aggregate increases by up to 60%. Shells surrounding nanolites of more evolved, higher viscosity melt[21,40,54,68] may then limit the rate of crystal growth, as the layers will be depleted in elements compatible with titanomagnetite crystallisation[29,52].

As the layer surrounding nanolites in our samples has a lower $\rho_e$ than the surrounding silicate glass and glassy sample MTL-G (Figs. 4, 5), it has a lower $\rho$, reflecting enrichment in Si and Al in the boundary layer, and depletion in Fe and Ti following crystallisation of titanomagnetite nanolites. However, observations of areas with an $\rho_e$ intermediate between that of the nanolites and silicate glass may also indicate the presence of ultrananolites with a size that is below the resolution of the technique. Due to the small size of the lapilli, fast quenching in air on ejection from the vent (seconds to a few minutes)[69] may have limited chemical diffusion across the boundary layer and post-fragmentation modifications, preserving the boundary layers.

## Effect on magma viscosity

To evaluate the rheological impact of the nano-particle bearing suspension, we used several rheological models and our 3D data to calculate the melt viscosity and the relative viscosity increase attributed to nanolite crystallisation (Supplementary Table 5), using the measured crystal volume fraction and aspect ratio ($r_p$). Below we present the results of calculations where the crystal-free melt viscosity is estimated using the Giordano et al.[70] model, however, we also compare our calculations with the more recent Langhammer et al.[71] model based on artificial neural networks. Further details on the viscosity calculations and estimations of uncertainties are provided in the Supplementary Information.

First, we used the Giordano et al.[70] model to quantify the crystal-free melt viscosity, using the MTL pre-eruptive conditions[7]. The calculated melt viscosity of the basaltic MTL composition is $6.1 \times 10^1$ Pa s at 1080 °C, using the average melt inclusion composition (Supplementary Table 1) and a $H_2O$ content of 2 wt.%, as measured in melt inclusions of the MTL eruption[7]. Considering the model uncertainty, the estimated melt viscosity ranges between $2.1 \times 10^1$ and $1.8 \times 10^2$ Pa s.

Due to the chemical modification of the melt following the crystallisation of titanomagnetite and potentially clinopyroxene nanolites, the glass surrounding nanolites (Fig. 2a-d) is trachy-andesitic in composition (Supplementary Table 1)[8]. The estimated melt viscosity is $3.2 \times 10^2$ Pa s at 1080 °C, ranging between $1.1 \times 10^2$ and $9.2 \times 10^2$ Pa s considering the model uncertainty[70]. Nanolite crystallisation can, therefore, increase melt viscosity due to compositional changes in the residual melt, by decreasing FeO and increasing $SiO_2$. However, recent studies show that viscosity models may overestimate the melt viscosity of liquids, as nanolites can crystallise over laboratory timescales during viscosity measurements in melts prone to nanocrystallisation[21,35-38], and may form a heterogeneous distribution of chemical components at the nanoscale[40]. Therefore, our calculations of the viscosity increase due to the changes in melt composition following nanocrystallisation may also be an underestimate.

Nanolites will increase magma viscosity due to their presence as solid particles. We used the Vona et al.[72] model and the equation of Costa et al.[73] to calculate the relative increase in magma viscosity due to the presence of crystals. Incorporating the measured nanolite crystal fraction ($0.27 \pm 0.02$) increases magma viscosity to $1.2 \times 10^4$ Pa s for the trachy-andesitic composition. Overall, considering both the effect on melt composition and contribution as solid particles (Fig. 7), nanolite crystallisation may increase magma viscosity by a factor of 20-200 (Supplementary Table 5).

Experiments demonstrate that the interaction between particles, and, consequently, magma viscosity is also strongly influenced by particle shape[74], due to the dependency of the maximum packing fraction ($\Phi_m$) and, therefore, the onset of non-Newtonian behaviour, on $r_p$[14,72,74,75]. As our ptychographic images provide insight into the 3D morphology of nanolites, we can use these data to calculate $r_p$. From our 3D images, we observe that the measured $r_p$ for the nanolites ranges between 1 and 4 (Supplementary Tables 2-3). The flow index describes shear-thinning behaviour and depends on both $\Phi/\Phi_m$ and $r_p$[14], where the transition between Newtonian and non-Newtonian behaviour occurs at a flow index of 0.9. At the measured $\Phi$, the higher values of $r_p$ may lead to the onset of non-Newtonian behaviour (Fig. 8).

Finally, the impact of nanolites on the rheology of suspensions is also dependent on their size and $N_a$. The $N_a$ of nanolites is considerably higher than microlites for the same crystal fraction, reducing the interparticle distance, affecting the local flow of interstitial melt, its viscosity and promoting non-Newtonian behaviour, even at low nanolite contents[13]. Nanolites may increase magma viscosity by up to two orders of magnitude[21,35,37], even at low particle contents of <5 vol.%[13,36,40].

Using the relative viscosity increase measured from the results of analogue experiments on nano-particle bearing suspensions, the magma viscosity of Etna trachy-basalt was estimated to increase from $10^2$ to $10^6$ Pa s with a nanolite content of 30%, calculated at pre-eruptive conditions[13]. At higher strain rates of $1 s^{-1}$, as estimated for basaltic Plinian eruptions[9], this viscosity increase occurs with a lower nanolite content of 10%[13]. Instead, comparing the results of analogue experiments using $SiO_2$ microspheres of a similar shape and at strain rates of $1 s^{-1}$, the same magma viscosity is reached with a higher microlite content of 60%[13,14,74]. As our melt composition is comparable to the trachy-basaltic Etna composition examined by Di Genova et al.[13], we can also suggest that the addition of the nanolite content estimated for the MTL magma (~30%) may have resulted in a comparable increase in the relative magma viscosity.

## Implications for eruption dynamics

Explosive eruptions arise due to magma fragmentation during ascent, a process which depends on magma viscosity, the timescale of deformation and the timescale of melt structural relaxation[3]. Nanolite crystallisation may occur during magma ascent[24,25,30,37], leading to significant changes in magma rheology during ascent and transitions in eruptive style[13,24,26-28,30,32,39].

The FL and MTL scoria lapilli were collected from proximal sampling localities[6-8] and likely experienced minimal post-fragmentation modification following eruption. Their small size (0.3-3 cm in

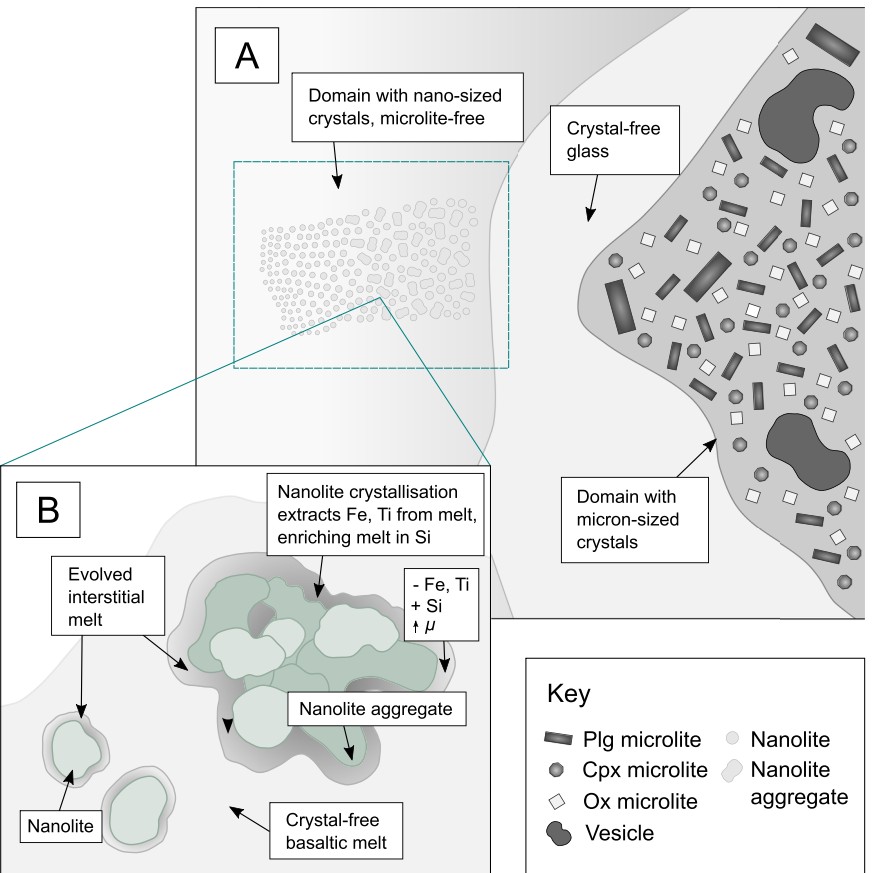

**Fig. 7 | A diagram illustrating the crystallisation of nanolites, the formation of a nanolite aggregate and the impact on melt viscosity.** Shown are **A** crystal-rich and crystal-poor domains observed in a basaltic scoria clast, where the microlite-free domains appear to show evidence of nanolite crystallisation. Within these domains, **B** nanolites crystallise, also forming aggregates. The crystallisation of Fe-Ti oxide nanolites leads to the formation of Fe and Ti-depleted melt surrounding nanolites, which is enriched in Si. This compositional boundary layer surrounding nanolites is also of higher viscosity ($\mu$) than the surrounding crystal-free basaltic melt.

diameter) will have allowed rapid cooling within the atmosphere following ejection, minimising post-fragmentation crystallisation. The micro-textures observed within the lapilli may represent features which developed prior to magma fragmentation and be used to interpret conduit processes. This is evidenced by the textures observed in the BSE images (Fig. 2a–d), which illustrate complex textures with irregular, sinuous contacts, indicative that the two magmas were potentially liquid upon emplacement, crystallising nanolites during mingling. The association between nanolites and vesicles (Fig. 2e, f) may indicate that nanolite crystallisation and vesiculation occurred at a similar time. Rhyolite-MELTS simulations indicate that the measured nanolite crystal fraction can be reached at 1040 °C and 10 MPa, likely within the shallow volcanic conduit during magma ascent. The $\rho_e$ distribution (Fig. 4) also indicates that nanolites crystallised in situ from the surrounding melt, due to the occurrence of haloes surrounding nanolites with a lower $\rho_e$.

The in situ observations of Di Genova et al.[13] demonstrate that nanolite crystallisation can occur on rapid timescales, with nanolites reaching 8 nm in diameter within 2–3 min. In these experiments, nanolites crystallised at temperatures above the glass transition temperature ($T_g$) at high undercooling ($\Delta T = 40–250$ °C), consistent with the conditions estimated for rapid magma ascent within the conduit during basaltic Plinian eruptions[12,13]. Our STEM images show nanolites approximately 20–30 nm in diameter, which may have crystallised within a few minutes. Instead, aggregates will have crystallised over a longer timescale. This timescale is consistent with the rapid magma ascent time estimated for the MTL eruption from numerical simulations of 1–13 min[8].

The FL and MTL Plinian eruptions represent unusually highly explosive events at basaltic systems. As crystal-poor basaltic magma is of low viscosity, gas-melt decoupling can occur during ascent, preventing magma fragmentation and promoting effusive, or less intense explosive activity. The often highly crystalline groundmass of samples of basaltic Plinian activity suggests that rapid microlite crystallisation may have an important role in increasing magma viscosity, promoting fragmentation and an explosive eruption, alongside rapid bubble nucleation and magma ascent[6–8,12,16–19,76]. Previous work[7] which analysed micron-scale textures in MTL scoriae found that rapid syn-eruptive microlite crystallisation may have increased the effective magma viscosity to $10^6$ Pa s, increasing the potential of magma fragmentation during ascent. Syn-eruptive microlite crystallisation can provide sites favourable for the heterogeneous nucleation of bubbles[15], whilst the rigid crystalline network can increase the tortuosity of connecting channels between bubbles, increasing gas-melt coupling and restricting outgassing[18,76].

FL and MTL clasts show considerable textural heterogeneity at the micron-scale (Fig. 2), where crystal fraction varies between 0.05 and 0.63[7,8,19]. The results of this study show that crystal-poor regions may also be crystallised at the nanoscale, however, in this case, the crystal size is below the resolution of the analytical technique, producing the mottled texture observed in high-resolution BSE images (Fig. 2b, d). This mottled texture has also been observed in the groundmass of clasts of the 1886 basaltic Plinian eruption of Tarawera, suggestive of incipient crystallisation[17]. Previous work found that the estimated magma viscosity calculated using the micron-scale crystal content is $10^2$ Pa s for crystal-poor regions of the MTL samples, whilst crystal-rich

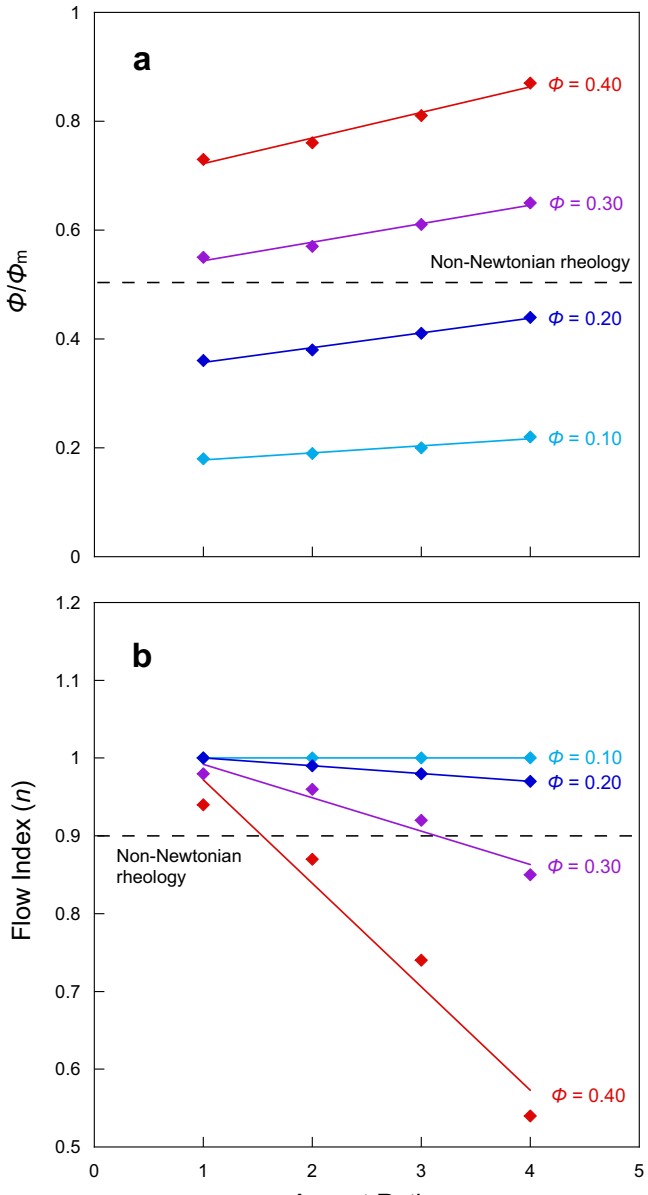

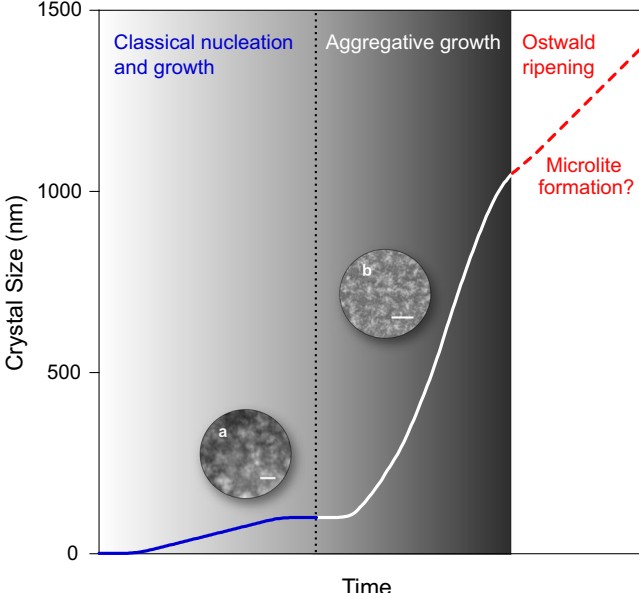

**Fig. 9 | Schematic representation of nanolite growth mechanisms.** Small (<50 nm), subspherical-spherical nanolites, such as those observed in the STEM images, may form through a classical nucleation and growth mechanism. With increasing time, nanolites may aggregate to form larger, more irregular shaped structures such as those observed in the ptychographic images. The textures observed in the samples show less evidence of Ostwald ripening. Instead, Ostwald ripening may lead to the formation of microlites. Inset **a** shows a HAADF-STEM image (Fig. 3d) of spherical-subspherical nanolites (scale bar = 25 nm), whilst inset **b** shows a SEM image (Fig. 2b) of larger aggregates (scale bar = 2 μm).

**Fig. 8 | Impact of aspect ratio on the onset of non-Newtonian rheological behaviour. a** Aspect ratio vs $\Phi/\Phi_m$, where $\Phi$ is the crystal fraction and $\Phi_m$ is the maximum packing fraction. **b** Aspect ratio vs the flow index ($n$), where the onset of non-Newtonian behaviour occurs at $n \leq 0.9$, which occurs at higher values of $\Phi$ and the aspect ratio.

domains have a higher estimated viscosity of $10^6$ Pa s, due to the increased microlite content[7]. However, our estimation of the viscosity of the nanolite-bearing crystal-poor glass ranges between $1.4 \times 10^3$ to $1.2 \times 10^4$ Pa s when the effects of nanolite crystallisation are considered. We observe that nanolite crystallisation can further increase magma viscosity through aggregation and the influence on melt composition and structure. Therefore, the rheological impact of apparently crystal-poor glass may be underestimated by considering only the micron-scale crystal fraction (Fig. 7). For magma compositions that are prone to nanocrystallisation[21], such as the tholeiitic basaltic magmas erupted at Las Sierras-Masaya volcanic system, assessment of whether the crystal-poor glass is nanolite-free may be important for determining the rheological impact of crystallisation and estimating bulk magma viscosity. Alongside syn-eruptive microlite crystallisation and dis-equilibrium degassing[7–9,12,16–19,76], nanocrystallisation would further

increase the viscosity of apparently crystal-poor melt[13,24,25,28,36,37], increasing the potential of magma fragmentation and a highly explosive basaltic eruption.

## Aggregation of nanocrystals

Aggregates may have a greater impact on magma rheology[13,77], and in turn, influence the potential of magma fragmentation within the conduit. Due to the large surface area of nanoscale crystals, nanolites are prone to aggregation through van der Waals interactions[13,41]. Collisions between nanolites due to Brownian agitation can facilitate nanocrystallisation and aggregation, leading to the formation of structures 100 s of nm in size[13,41]. In volcanic conduits, nanolites may interact with each other and aggregate under the influence of shear flow[34].

Our 3D ptychographic images provide insight into the interaction between nanolites and the formation and morphology of aggregates. Where nanolites are aggregated, they form structures (Fig. 6) ranging between 400 nm and 6 μm in size (Supplementary Tables 2, 3), comparable with the elongate, aggregated structures observed in 2D images (Fig. 3e, h)[30]. This aggregation will increase the effective volume of nanolites[13,55].

Nano-sized crystals may grow through several mechanisms, such as classical nucleation and growth, aggregation and Ostwald ripening[78]. All three processes can occur sequentially through time (Fig. 9), contributing to the final texture and nanolite distribution[78,79]. The small (20–30 nm) spherical-subspherical nanolites observed in the STEM images may reflect the classical mechanism of crystal nucleation and growth. However, larger, irregular-shaped nanolites may result from particle aggregation and/or Ostwald ripening.

Ostwald ripening describes the growth of larger crystals from the dissolution of smaller ones, reducing $N_a$ and increasing the mean crystal size whilst maintaining the volume fraction[78,80]. Particle morphology may be used to distinguish between Ostwald ripening and aggregation[78,81]. In situ TEM observations of Au and Pt nanocrystal

growth show that in simple growth, a single particle grows whilst maintaining a nearly spherical shape[79,81]. Particle interactions and coalescence then produce polycrystalline particles with a more irregular morphology[79,81]. In our images (Figs. 2–4) we observe spherical-subspherical titanomagnetite crystals and more elongate, irregular shapes, instead of the euhedral crystals which may be expected from Ostwald ripening[80]. Therefore, Ostwald ripening likely did not occur, possibly due to the short timescale of crystal growth. In silicate systems, experiments show euhedral olivine and plagioclase can result from Ostwald ripening, over long timescales and at low undercooling[80]. Instead, SAXS and WAXS experiments indicate that nanoscale aggregates may form within minutes in a basaltic composition[13].

In igneous rocks, crystal aggregates can form from mechanisms attributed to crystal growth or dynamic processes[49,55,72,82,83]. Synneusis is a hydrodynamic process that can occur in convecting magma or during settling, leading to the attachment of previously isolated crystals[83,84]. During synneusis, melt films can become trapped between the attached crystals, which often have different core compositions[83]. Conversely, aggregates which form from processes related to crystal growth, such as epitaxy, heterogeneous nucleation or dendritic growth, will comprise of crystals with a similar composition in crystallographic orientations related to the main crystal host[55,82,83]. Our ptychographic images illustrate that the interstitial melt between nanocrystals represents a boundary layer related to their crystallisation[52,55].

Heterogeneous nucleation and epitaxial growth of titanomagnetite on clinopyroxene has been observed to form crystal aggregates in a synthetic Fe-rich basalt[82]. In epitaxial growth, growth of the crystallising phase occurs preferentially on surfaces with a specific crystallographic orientation[82]. Although distinguishing between these mechanisms requires detailed information on crystallographic orientation at the nanoscale, from our multi-scale imaging we can suggest that the observed titanomagnetite aggregates may originate from a mechanism related to crystal growth rather than a dynamic process.

In conclusion, we present cutting-edge 3D observations of nanolites in the glass of scoriae from highly explosive basaltic eruptions, obtained using X-ray ptychography. Our results provide an innovative, direct visualisation of nanolite aggregation in natural volcanic samples in 3D, a phenomenon described from the results of experiments and modelling. Titanomagnetite nanolites likely extract Fe and Ti from the melt during crystallisation, developing layers of differentiated, Si-rich melt around nanolites. The $\rho_e$ distribution observed in the 3D ptychographic images shows nanoscale heterogeneities in glass composition and the development of differentiated boundary layers surrounding nanolites.

Our 3D images show that aggregation of individual nanolites (20–300 nm) can produce aggregates that range between 400 nm and 6 μm in size. Growth of the individual nanolites likely follows a classical mechanism of crystal nucleation and growth. Instead, aggregates may form through a mechanism related to crystal growth such as epitaxy, heterogeneous nucleation or dendritic growth, rather than a dynamic process. Evidence for Ostwald ripening is not observed at this scale and may instead lead to the formation of microlites.

Nanolites can, therefore, increase magma viscosity through a combination of their presence as solid particles in the magma, their interaction and aggregation and the chemical and structural modification of the melt induced by their crystallisation. The presence of nanolites in the scoriae of the FL and MTL basaltic Plinian eruptions suggests that the syn-eruptive crystallisation of nanolites during magma ascent may increase magma viscosity and the potential for magma fragmentation, leading to a highly explosive eruption.

## Methods
### Sampling
The Fontana Lapilli (FL) eruption of Las Sierras-Masaya volcanic system (60 ka) erupted 2.9–3.8 km³ of tephra during the main Plinian phase, reaching a maximum plume height of 32 km[85]. The mass eruption rate

of the Plinian phase is estimated as $1.4 \times 10^8$ kg s⁻¹. Following the stratigraphic framework of Costantini et al.[85], the deposit is divided into 8 units (A-G), representing a period of continuous deposition over the duration of the eruption (estimated as 4–6 h). Our FL samples are scoriae 1–3 cm in size and are from the opening (Unit A/B) and Plinian phases (Units D-LG), where FonE refers to unit E of the FL eruption[8,19]. The sampling locality is that of Bamber et al.[8].

The more recent Masaya Triple Layer (MTL) eruption (2.1 ka) ejected 3.4 km³ of tephra, forming an eruption column with an estimated height between 21–32 km, with episodes of phreatomagmatic activity[6]. The maximum eruption rate is estimated as $1 \times 10^8$ kg s⁻¹. Scoriae from the MTL eruption were obtained from Plinian fallout layer TLL2 and are scoriae 0.3–2 cm in size. The sampling locality is the same locality which is described in Bamber et al.[7].

### 2D imaging techniques
High-resolution BSE images of scoriae were obtained using the JEOL JXA-8530F FEG electron microprobe at the Photon Science Institute, University of Manchester. An acceleration voltage of 15 kV and a beam current of 7–10 nA was used during acquisition. Quantitative textural analysis was performed using 2D images (Fig. 2b, d) and the software ImageJ[86], to estimate the nanolite crystal fraction over a larger area (Supplementary Fig. 2).

The high annular dark-field (HAADF) STEM micrographs were obtained using the FEI Talos scanning transmission electron microscope (STEM) at the Department of Materials, The University of Manchester, equipped with an energy dispersive X-ray spectrometer (EDS) for compositional analysis. Samples were prepared by grinding scoriae of the MTL eruption using an agate mortar and pestle, to produce a fine-grained powder consisting of fragments ~20 μm in diameter. The powder was then added to a 400 μm diameter copper-carbon mesh using a pipette in preparation for STEM analysis.

### Sample preparation for X-ray ptychography
Sample MTL-G was extracted from a crystal-poor area of glass within a clast of the MTL eruption using a FEI Nova 600i focused ion beam (FIB). First, the area was inspected using high resolution BSE images of the glass to check for the presence of microlites, obtained using the FEI Quanta 650 FEG-SEM at the Department of Earth and Environmental Sciences, The University of Manchester, using an acceleration voltage of 15 kV and a beam current of 10 nA. The sample was then extracted from this area of crystal-poor glass using the dual-beam FEI Nova 600i FIB at the Department of Materials, The University of Manchester, equipped with a Ga+ ion beam and an electron beam. The area was sputtered using the Ga+ ion beam to mill material from the glass and extract the sample, using an acceleration voltage of 30 kV and a beam current of 6.5 nA. BSE images were collected using the SEM throughout the milling process to monitor the extraction of the sample. The 10 μm cubic sample of glass was then extracted and mounted on a tungsten needle (Fig. 1b) in preparation for analysis using X-ray ptychography.

Samples MTL-A and MTL-B are ~30 μm individual fragments of scoriae, prepared as a fine powder by grinding scoriae in an agate mortar and pestle. The individual samples were then mounted on a carbon fibre (diameter = 7 μm) using a micro-manipulator. This fibre was then mounted on a wooden stick and placed on the rotation stage for acquisition using X-ray ptychography.

### X-ray ptychography
The experiment was performed using the ptychography end station of the I13-1 beamline at Diamond Light Source[51]. The experimental setup is shown in Fig. 1. The coherent X-ray is focused down to a 3 μm illumination at the sample using a set of blazed Fresnel zone plates (FZP in Fig. 1a)[87] (with a 400 μm diameter and 150 nm outer zone width). Only the first diffraction order of the FZP reaches the sample. The higher diffraction orders are blocked using an order sorting

aperture (OSA in Fig. 1a- 10 μm diameter iridium pinhole) located close to the focus of the first diffraction order. The zero order is blocked using a central stop (CS in Fig. 1a - 50 μm diameter, 50 μm thick gold) placed upstream of the FZP. The sample is mounted on the top of a 3-axis PI-Mars piezo stage which sits above a rotation stage. For acquiring the 3D ptychographic dataset, the sample is rotated over a range of 180 degrees, with a step size of 0.18 degree. At each angular position, the sample is scanned across the illumination in a 32 × 40 step grid, with a 1 μm step size, using the piezo stage. The diffraction patterns are recorded at each scanning position with a Merlin photon counting detector (silicon chip, 512 × 512 pixels, 55 μm pixel size).

Two different configurations were used during two experimental sessions. The first session used a 15 keV X-ray beam and a sample-to-detector distance of 4.2 m. This configuration was applied to image sample MTL-B. This configuration corresponds to a reconstructed pixel size of 14 nm. The second session used a 11.7 keV X-ray beam and a sample-to-detector distance of 8 m. This configuration was applied to image samples MTL-G and MTL-A. This configuration corresponds to a reconstructed pixel size of 60 nm. The image reconstruction was performed using 1000 iterations of the ePIE algorithm[88] implemented in PtyREX code[89]. The resolution of the instrument was measured during the experiment by imaging a Siemens star standard test resolution target (Supplementary Fig. 3) and was determined to be 50 nm.

The segmentation of nanolites from the glassy matrix and the segmentation of the evolved glass around and between nanolites was operated in the 3D domain using Avizo software (v.2019.1; Thermo Fisher Scientific, USA). Manual bi-level greyscale thresholding was applied, based on the greyscale histogram of the selected VOIs and visual inspection of the slices in different directions (axial, coronal, and sagittal). This approach follows that of Arzilli et al.[55]. The reconstructed segmented 3D images were then analysed using Avizo software, where the volumes and dimensions of nanolites and the glass around nanolites were measured using the label analysis module.

3D visualisation (volume renderings) of the reconstructed volumes was obtained using the commercial software VGStudio 3.0 (Volume Graphics). The volume renderings were used to perform 3D textural observations of nanolite morphology and the interaction between nanolites and the surrounding evolved glass (Fig. 6). The segmentation of the nanolites, surrounding evolved glass, and matrix glass was based on both the visual inspection of images and the quantitative information provided by the estimates of $\rho_e$. As each pixel in the ptychographic images provides information on $\rho_e$, and the nanolite and glass phases have a specific range in $\rho_e$ which reflects their composition (Table 1), the spatial variation in $\rho_e$ was also used for thresholding and the segmentation of each phase (Fig. 4). The glass control sample MTL-G (Fig. 5) was used as a reference for the $\rho_e$ of basaltic glass and for the segmentation of the matrix.

### Atomic density

To calculate the $\rho_a$ of the nanolites and the glass, the compositional data of Bamber et al.[7] were used, obtained from natural clasts of unit TLL2 of the MTL eruption. Probe for EPMA software (v.12.9.5) was used to calculate the average atomic number[60] for the crystal-poor basaltic glass ($n = 9$) and titanomagnetite ($n = 10$), where $n$ represents the number of EPMA analyses. The average atomic number estimated for the MTL basaltic glass is $12.91 \pm 0.07$ (1σ), and for titanomagnetite, it is $20.32 \pm 0.04$. These analyses were used to determine the reference value of $\rho_a$ for basaltic glass and titanomagnetite in natural clasts of unit TLL2 of the MTL eruption, for comparison with the unknown values of $\rho_a$ obtained for the nanolites and glass using the orthoslices.

### Data availability

The data generated in this study are provided in the Supplementary Information and have been deposited on Figshare under accession code https://doi.org/10.6084/m9.figshare.29430371.

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

## Acknowledgements

The work contained in this publication contains work conducted during a PhD study (E.C.B.) supported by the Natural Environment Research Council (NERC) EAO Doctoral Training Partnership and is fully funded by NERC, whose support is gratefully acknowledged (Grant number NE/L002469/1). The research leading to these results has received funding from the RCUK NERC Dis-Eqm project (NE/N018575/1) (M.R.B.). The beamtime on the I13-1 endstation was provided by Diamond Light Source (Proposal number: MG23863-1) and laboratory space by the Research Complex at Harwell. E.C.B. acknowledges a postdoctoral fellowship ("Assegno di Ricerca") cofunded by the University of Turin and the LMU Munich. F.A. thanks the KINETICVOLC project funded by the European Union, NextGenerationEU, Ministry of University and Research, Promotion and Development Fund DM737/2021. F.A. and D.D.G. thank the CRYSTALKIN project funded by the PRIN2022 (2022L2APNR), Next-GenerationEU, Ministry of University and Research. M.P. acknowledges funding from UKRI FLF project 4DVOLC (MR/V023985/1). D.D.G. acknowledges the funding from the European Research Council (ERC) under the European Union's Horizon 2020 research and innovation programme (NANOVOLC, ERC Consolidator Grant–No. 101044772). D.D.G. and P.V. acknowledge the funding by Deutsche Forschungsgemeinschaft (DFG) project DI 2751/2–1.

## Author contributions

E.C.B. conceived the research project, with F.A., G.L.S., M.P., D.D.G., and M.R.B. E.C.B. collected the BSE images and performed the HAADF-STEM analysis with H.B. The 2D images were analysed by E.C.B. and F.A. with contributions from H.B., R.B., and D.D.G. A.G. prepared the samples for X-ray ptychography using FIB. The X-ray ptychography experiments were performed by E.C.B., F.A., S.C., D.J.B., G.L.S., M.P., and M.R.B. The X-ray ptychography data were processed by E.C.B., F.A., S.C., and D.J.B. The viscosity calculations were performed by E.C.B., F.A., D.D.G., D.G., and P.V. All authors contributed to the production of the manuscript.

## Competing interests

The authors declare no competing interests.
