## [Transparent Peer Review file · Nature Communications]

3D quantification of nanolites using X-ray ptychography reveals syn-eruptive nanocrystallisation impacts magma rheology

Corresponding Author: Dr Emily Bamber

Version 0:

Reviewer comments:

Reviewer #2

(Remarks to the Author)

The key results of this manuscript are that abundant nanolites (crystals $<1 \mu\text{m}$ in size) and clusters of nanolites (up to $\sim 6 \mu\text{m}$ in size) have been imaged in scoria from a basaltic Plinian eruption deposit in Nicaragua. The X-ray ptychography technique used to image the nanolites is exciting and seems to have great potential. The agglomeration process and volume of interstitial melt that can be trapped is discussed, although some of the arguments need clarification. The compositions of the uncrystallized glass, and glass around the nanolites, are used to estimate rheological changes due to crystallization, and due to agglomeration of nanolites. The effects of crystallization on magma rheology and potentially crossing the ductile-brittle transition are undoubtedly very important, although the effects of agglomeration are less convincingly presented. In this section it would be very helpful to explicitly mention the uncertainties in all the models used, from melt viscosity to the effect of crystals, and all the assumptions being made, for example temperature, water content (which should change during crystallization, along with the rest of the melt composition), strain rate, etc.

In summary, the results are interesting and add to a growing body of literature suggesting that nanolites must be considered along with microlites when considering evolving magma rheology. This is important in volcanology, and especially to understanding basaltic Plinian eruptions. Comments below are divided into 5 main themes. I hope they will be useful in revising the manuscript.

[1] Please try to be specific when citing references.

For example on line 88, not every reference in the range 19-37 is about nanolites, so there should be some selection applied and consequent re-ordering of the references. Another example is on line 102, where reference 43 is about nanofluid viscosity but it does not mention either agglomeration or van der Waals forces, so this is not the best place to cite it. Please just double check that the appropriate references are cited in each place throughout the manuscript.

[2] The nature of the chemical zoning in the melt around the nanolites deserves closer scrutiny and possibly a different description.

e.g. On lines 204-212 it is asserted that the halo has “an p_e intermediate between that of the silicate glass (Fig. 4) and nanolites”. In order to get a melt that was enriched in Fe and Ti relative to the uncrystallized glass would imply either (i) it could contain ultrananolites that are smaller than the spatial resolution of the technique, or (ii) some previously formed nanolites had dissolved into the melt, which would require some explanation.

But some of the glass next to the nanolites actually has a lower p_e than the background glass – this is most easily seen by colour-inverting Fig 4a. Look just left of center, and also towards the bottom left hand corner. This makes much more logical sense – if Fe and Ti have been removed from the melt layer from which titanomagnetite nanolites formed, that residual melt should be less dense (consider for example the partial molar volume data of Lange and Carmichael 1987 CMP and more recent studies, which suggest that FeO and TiO₂ are the densest major oxides while SiO₂ and Al₂O₃ are much less dense). The reason this is important is that if the nanolites formed in situ, i.e. from the melt with which they are currently associated, this is the pattern one would expect. If they have migrated through the melt they may have been able to shed their FeTi-depleted halo, which one would expect to lead to different properties of the bulk magma.

e.g. On lines 263-270 it is again suggested that “the glass in the nanolite-bearing samples is likely of higher ρ and a different composition” but the example from the literature used to justify this is one where the nanoparticles are surrounded by a “Si-rich ‘shell’ ($\rho = 2.2 \text{ g/cm}^3$)”. In that example, the matrix has a density of 2.7 g/cm^3 , so the Fe-depleted shell is lower density. There is no reason to think that is not also true for the samples presented here, as detailed in the previous paragraphs. For the shell to be higher density than the surrounding glass would require some remarkable geochemical behaviour that would require additional explanation.

e.g. Line 305. Normal-sized crystals, which are also surrounded by haloes of depleted melt (boundary layers), still manage to grow. So there may be additional justification needed to assert that the halo prevents further growth of nanolites. And does “gas escape” here refer to actual bubbles (nanobubbles?) or to the diffusion of volatiles, which will also be enriched in the halo? This needs clarifying because there are structures in Figure 2 ascribed to the collapse of bubbles that have escaped. One important thing to note is that the halo is depleted in elements that formed the crystal, but enriched in the other elements relative to the adjacent melt reservoir. Chemical diffusion along this gradient will then drive homogenization, so one important question is the timescale of quenching – in the lab or in a Plinian eruption this can be quick, preserving the haloes.

e.g. lines 308 to 314. The presence of undetectable ultrananolites is one possibility, another is dissolution of previously formed nanolites (although why this would happen would need explaining). A third possibility is that the haloes are in fact less dense than the surrounding melt – as appears to be the case from Fig. 4 – although that would appear inconsistent with the densities calculated in Table 1.

Lines 413 to 416. The melt compositions in Supplemental Table 3 also require cpx crystallization, but this should be mentioned explicitly earlier on, when reporting the melt compositions and attempts to identify the nanolite minerals. Is it possible that the phase contrast between clinopyroxene nanolites and residual glass would be so small as to be essentially undetectable with this technique? Could this potentially resolve the problem of a dense melt halo (lines 204-212)?

[3] The discussion of the effects of agglomeration probably needs some refining.

e.g. Line 133 how does agglomeration of existing particles produce “chemical changes in the surrounding melt which markedly increases their overall contribution to magma viscosity”? Nucleation and growth of new particles would cause changes in the surrounding melt due to depletion of Fe and Ti, but is there a reason for agglomeration of existing particles (presumably with their own depleted haloes) to cause further changes?

e.g. lines 326 to 328. “... agglomeration will increase the effective volume of nanolites. Comparing the segmentation of the nanoscale aggregates both with, and without, the incorporation of the surrounding glass (Fig. 6), the initial volume of the aggregate increases by up to 60 %.”

But Fig 6 is comparing the agglomerate with and without its depleted melt halo. The proper comparison is between the agglomerates and the sum of the individual particles with their individual haloes. Agglomeration could result in a smaller volume of enriched melt, because the agglomerated particles presumably have a smaller total surface area than the sum of the individual particles. If the depleted halo compensates for this by thickening, then perhaps there is simply no change in its effective volume.

The effective volume could increase if additional (undepleted) melt were trapped between the particles due to agglomeration. For example, random close packing of uniform spheres yields ~36% space between the spheres. So perhaps the effective volume could be 36% higher, but if the particles need to touch to stick (as van der Waals forces decay extremely rapidly with increasing distance), much of the interstitial melt is likely the depleted halo material.

In short, if a clearer way can be found to explain how agglomeration leads to an effective volume increase, and what the quantitative magnitude of that increase could be, it would be appreciated.

[4] The discussion of calculated viscosities needs careful inclusion of uncertainties and better explanation of the assumptions being made

Lines 348 to 352 – please give the model uncertainties on these calculations. For GRD the RMSE for volatile-bearing melts is ± 0.46 log units (their Table 3) so combining that with the ± 0.17 log unit RMSE for Langhammer et al. (2021) means the calculated melt viscosity is 9-19 Pas (Langhammer, best estimate 13 Pas with ± 0.17 log unit rmse) or 21-176 Pas (Giordano with ± 0.46 log units rmse). Using the ± 0.74 log unit RMSE for the GRD model applied to the Langhammer et al. dataset, gives a range of 11-335 Pas, which is already a substantial overlap with the Langhammer model. Given that RMSE is a one sigma uncertainty, the two models in fact agree quite well, even though that gives a total range of 9-335 Pas (or 0.9-2.5 log units). I do agree that it is useful to show how much uncertainty there still is even for predicting melt viscosity.

Line 350 – why assume 2 wt.% H₂O – how was this water content determined (by previous studies)? For the evolved melt composition, are you also using 2 wt% H₂O? Crystallization of 27 vol.% crystals should increase the water content in the melt by ~35%, and this should be factored into the magma viscosity calculations.

Lines 357-361 I think the text is saying that all existing viscosity measurements (and hence models) are suspect because nanolites may have affected them. For the specific point being made, that would likely affect measurements of both the initial liquid and the depleted halo liquid – there is still plenty of Fe and Ti in the halo (Supp Table 3). But it is also worth remembering that crystals will not form below the liquidus, and that fast cooling rates seem to be required to form nanolites rather than microlites – which are easily detected by microscopic inspection. Nanolites form at cooling rates of several °C/s (Di Genova et al. 2020) but viscosity measurements by e.g. micropenetration or parallel-plate viscometry typically use much lower cooling rates. And even if they were to form, a sudden nanolite nucleation event should be easily detected in the

viscosity data as a discrete jump, given the strong effect they can have on viscosity. So on balance this argument does not feel well justified.

Line 363 please provide the actual equation for relative viscosity as a function of crystallinity that was used here. Which of the approaches used in Vona et al. (2011) was used here? These include a modified Krieger-Dougherty equation from Ishibashi (2009), the Costa et al. (2009) model, and then a new parameterization based on crystal aspect ratio and strain rate. All three of these are strain-rate-dependent and so the strain rate used in the calculations needs to be stated.

Line 373 To summarize, the calculated evolved melt viscosity is 35 Pas (actually 24-52 Pas allowing for ± 1 rmse from Langhammer) or 320 Pas (actually 110-920 Pas allowing for ± 1 rmse from Giordano). The calculated magma viscosity is 150-1400 Pas (Mader model) or 1400-12000 Pas (Vona model). The estimated increase is then a factor of ~ 4 -5 (Mader) or 40-50 (Vona), but not 10-100. Bearing in mind the effect of increased water content in the evolved melt, which is not considered here, the actual viscosity change is likely to be less than this. It is still obviously an important change, but the way things are written seems to always be pushing the limits of the maximum effect that could be claimed, while ignoring the uncertainties in all of these models.

Line 380 there needs to be additional context on the basalt with 30% nanolites reaching a viscosity of 10^6 Pas. What temperature, composition, water content (if any) and what would the crystal-free melt viscosity be at those conditions? Likewise additional context is needed for stating that 60% microcrystals are needed to achieve the same viscosity as 30% nanolites. A quick look at e.g. Figure 2 of Costa et al. (2009) indicates there is a huge uncertainty in the experimentally determined effect of micro-crystallinity on effective magma viscosity. Lejeune and Richet (1995) documented a relative viscosity of more than 1000 at only 40% crystals in one experiment, so at the very least there needs to be acknowledgement of the uncertainties in these statements.

Line 384 relative viscosity increase is mentioned here but the starting viscosity is not – what was the relative viscosity increase documented by Di Genova et al., and what actual viscosity would that imply starting with the melt compositions and viscosities discussed above?

Lines 393-397 “The micro-textures observed within the lapilli may, therefore, represent features which developed prior to magma fragmentation”.

It is tempting to assume that the crystal textures all formed pre-fragmentation, but wouldn't these clasts have cooled (post-fragmentation) at almost exactly the rates used to stimulate nanolite crystallization in previous experiments such as by Di Genova et al. (2020)? An estimate of the cooling rate could be made based on the pyroclast thermal model of Moitra et al. (2018 G-cubed). For 2-3 cm clasts the cooling rate should be $\sim 4^\circ\text{C/s}$ from 1000 to 600°C . Processes in the conduit prior to fragmentation would likely involve much slower cooling, at ranges typically associated with microlite growth. This should perhaps be stated more clearly.

Alternatively, if the timescale of “a few minutes” applies to both nanolite growth and ascent to the surface this implies that magma mixing caused an immediate eruption (with a delay of less than a few minutes). How could these two magmas mix that quickly? Do they have different compositions? Were they stored at different temperatures? If the answer to either or both of these is yes, the rheological calculations should compare the two magmas. Such rapid eruption triggering would imply a sudden degassing event, but what exactly about the mixing process would have caused it?

Lines 423-425. The assertion that the MTL magma viscosity did reach 10^6 Pas in the conduit, from Bamber et al. (2020), appears based on a microlite fraction of 0.5 and a starting melt viscosity of 10 Pas (section 4.3.2) or 30-40 Pas (section 4.2.3), and the Costa et al. (2009) model. Costa et al. (2009) present a framework for fitting experimental data but not a specific set of recommended parameters. Mader et al. (2013) suggested a set of parameters for use with Costa et al. (2009) but they are somewhat arbitrary – the data in Fig 2 of Costa et al. (2009) show that there is no single equation (or parameters) that can describe relative viscosity as a function of crystal fraction. The MTL magma viscosity quite possibly could have reached 10^6 Pas, but again there are uncertainties that need to be acknowledged.

[5] Miscellaneous minor points:

Line 1 The title might be more accurately written as “3D visualisation of nanolite aggregates in basaltic magmas...” because the process of aggregation is not seen in a static image.

Line 64 “high viscosity facilitates the accumulation of stress due to applied strain”. This may need rephrasing – in volcanic systems the stress is applied and the strain results from that, not vice-versa. High viscosity means a low strain rate and hence low rate of stress dissipation, but it does not cause the high stress in the first place.

Line 74 how does “fracturing ... modify the rheology of basaltic magma”? Isn't fracturing an indication that the rheology has already changed from the ductile to brittle regime, through some other mechanism (e.g. high strain rate, change in effective viscosity, etc)?

Lines 90-92 is it necessary to list “increasing melt polymerization” separately from “chemical and structural changes in the residual melt”? It may be better worded as “by facilitating heterogeneous bubble nucleation or inducing chemical and structural changes in the residual melt, for example increasing melt polymerisation, ...”

Lines 176-178 I agree that there does not appear to be Si enrichment around the nanolites – and for Fig 3m there does not appear to be much Si depletion at the site of the nanolites either, which is surprising.

Lines 223-225 and caption to figure 6. The caption should also explain why this part of the glass is shown and how it is distinguished from the rest of the glass (lower density). Where we can see the nanolites in Fig 6 c and d, does this imply a lack of the evolved boundary layer in these areas, or has it simply been cut away to reveal the nanolite inside? This is important to know. The 2D orthoslices suggest that the enriched boundary layer is not present everywhere around the nanolites.

Line 233 why is the unit of ρ_e cm⁻³ and not m⁻³? cgs units should be replaced with SI units wherever possible.

Line 242 does ρ_a have a name (e.g. is it the atomic density), and are the units m⁻³?

Lines 297-301 Si⁴⁺ is also a cation, so the boundary layers are not “depleted in cations” but they are often depleted in Fe and Mg or Ti. The key is that the boundary layer is enriched in network-forming Si⁴⁺ and Al³⁺ (don't forget the alumina!). The long sentence on lines 297-301 mentions viscosity increase in two different places, and should be restructured.

Lines 301-304. If you observe shells are 10-350nm in size then that is the size they can reach (at least!). Remove “can reach 10-100 nm in size” on line 301.

Line 341 “Nanolites can increase magma viscosity by up to two orders of magnitude”. So can larger crystals - it all depends on the volume fraction, which should be specified here. Unless this is meant to imply an additional increase relative to microcrystals at the same volume fraction, in which case that should be specified clearly.

Line 480 Should the beam diameter be 10 nm rather than 10 microns for the BSE imaging? A 10 micron diameter would be good for a glass microprobe analysis but not for high resolution imaging.

Lines 561-562 should the average atomic numbers be given as 12.91 ± 0.07 and 20.32 ± 0.04 ? Please specify whether the \pm is one standard deviation or two.

Fig. 2 is described in the caption as showing two pieces of scoria from the FL eruption, and the image labels are “FonAB” and “FonE”. These appear to be two different layers of the FL eruption but this should be clarified. Moving the sample descriptions from supplemental material into the main text would help with this.

Fig. 5 it would be helpful to see MTG at the same spatial scale as MTL-A is shown in Fig. 4 (as they were collected using the same geometry and resolution). To help make the comparison described in the text, this could become a third panel of Fig. 4, and the greyscale should be the same in all three panels (currently the greyscale range in Fig 5 maps to a different range of electron densities compared that used in Fig 4). Please also explain the cause of the “thumbprint” pattern evident in Figure 5 – is this some kind of artifact arising from the ptychographic technique or the image processing?

For the supplemental information, it would be helpful for the reader to have the sample information in the main article text. Supplementary Tables 3 and 4 would also be useful in the main text, as the glass compositions and calculated viscosities are discussed extensively.

Reviewer #3

(Remarks to the Author)

Review of manuscript NCOMMS-24-72256

Titled: “3D visualisation of nanolite aggregation in basaltic magmas using X-ray ptychography: Implications for magma rheology”

submitted for publication in the Journal of Volcanology and Geothermal Research.

Authors: Emily C. Bamber, Fabio Arzilli, Silvia Cipiccia, Darren J. Batey, Giuseppe La Spina, Margherita Polacci, Ali Gholinia, Heath Bagshaw, Danilo Di Genova, Richard Brooker, Daniele Giordano, Pedro Valdivia, and Mike R. Burton

Dear Dr. Neely,

Thank you for the opportunity to review this article. The manuscript presents a new approach to volcanic material characterization to provide a careful assessment of the Nanolite content using two explosive basaltic eruptions from the eruption of Las Sierras-Masaya volcanic system as case studies.

Measuring and characterizing Nanolite size and geometry is something that traditional volcanological studies have not done and this technique is a promising step toward understanding more about whether and how important these particles are in terms of determining eruption intensity and style. The presented multipronged approach, using a wide range of complementary analytical techniques is exciting and may help advance our understanding of these phenomena. The presented data and analysis are sound and support the main arguments.

In the annotated pdfs of the main manuscript and supplements I have attached to this letter I provide a range of comments that include minor suggestions for re-structuring and re-wording and some more in-depth analyses / questions about what could be discussed on the basis of these data. I realize that some of those may go beyond what the authors intend to present in this paper but wanted to include them in case there was capacity to address them here or in future studies. I also provide a few suggestions for where and how this work could be improved and where more detail / better reasoning may help. I hope this can serve to further improve this piece of research. Overall, I suggest minor revisions to address relevant comments

before publication.

Best Regards,
Stephan Kolzenburg

Version 1:

Reviewer comments:

Reviewer #2

(Remarks to the Author)

The authors have thoroughly revised the manuscript in response to reviewer comments and I now recommend publication.

Reviewer #3

(Remarks to the Author)

The rebuttal clearly states that comments raised by myself and other reviewers were considered in detail and the manuscript amended accordingly. I see no further need for revisions and suggest the manuscript be published in the resubmitted form.

We thank both of the reviewers for their constructive and insightful comments which have greatly improved our work. Reported below is a point-by-point response to each comment raised by the reviewers, with line references to where the corresponding changes can be found in the revised manuscript. These revisions are colour coded in the updated article file and the comments below, where responses to reviewer #2 are highlighted in blue and responses to reviewer #3 are highlighted in orange.

REVIEWER COMMENTS

Reviewer #2 (Remarks to the Author):

The key results of this manuscript are that abundant nanolites (crystals $<1 \mu\text{m}$ in size) and clusters of nanolites (up to $\sim 6\mu\text{m}$ in size) have been imaged in scoria from a basaltic Plinian eruption deposit in Nicaragua. The X-ray ptychography technique used to image the nanolites is exciting and seems to have great potential. The agglomeration process and volume of interstitial melt that can be trapped is discussed, although some of the arguments need clarification. The compositions of the uncrystallized glass, and glass around the nanolites, are used to estimate rheological changes due to crystallization, and due to agglomeration of nanolites. The effects of crystallization on magma rheology and potentially crossing the ductile-brittle transition are undoubtedly very important, although the effects of agglomeration are less convincingly presented. In this section it would be very helpful to explicitly mention the uncertainties in all the models used, from melt viscosity to the effect of crystals, and all the assumptions being made, for example temperature, water content (which should change during crystallization, along with the rest of the melt composition), strain rate, etc.

In summary, the results are interesting and add to a growing body of literature suggesting that nanolites must be considered along with microlites when considering evolving magma rheology. This is important in volcanology, and especially to understanding basaltic Plinian eruptions. Comments below are divided into 5 main themes. I hope they will be useful in revising the manuscript.

[1] Please try to be specific when citing references.

For example on line 88, not every reference in the range 19-37 is about nanolites, so there should be some selection applied and consequent re-ordering of the references. Another example is on line 102, where reference 43 is about nanofluid viscosity but it does not mention either agglomeration or van der Waals forces, so this is not the best place to cite it. Please just double check that the appropriate references are cited in each place throughout the manuscript.

We agree with the reviewer and have now revised the citations and references throughout the manuscript. We have made specific changes to the lines highlighted by the reviewer, which can now be found at lines 89 and 105 in the revised manuscript. We have also revised the citations in the discussion, particularly in the section on melt composition (line 304). Due to these changes we have also rearranged the references and restructured the reference list.

[2] The nature of the chemical zoning in the melt around the nanolites deserves closer scrutiny and possibly a different description.

e.g. On lines 204-212 it is asserted that the halo has “an ρ_e intermediate between that of the silicate glass (Fig. 4) and nanolites”. In order to get a melt that was enriched in Fe and Ti relative to the uncrystallized glass would imply either (i) it could contain ultrananolites that are smaller than the spatial resolution of the technique, or (ii) some previously formed nanolites had dissolved into the melt, which would require some explanation.

But some of the glass next to the nanolites actually has a lower ρ_e than the background glass – this is most easily seen by colour-inverting Fig 4a. Look just left of center, and also towards the bottom left hand corner. This makes much more logical sense – if Fe and Ti have been removed from the melt layer from which titanomagnetite nanolites formed, that residual melt should be less dense (consider for example the partial molar volume data of Lange and Carmichael 1987 CMP and more recent studies, which suggest that FeO and TiO₂ are the densest major oxides while SiO₂ and Al₂O₃ are much less dense).

We thank the reviewer for this comment and agree that the areas of glass adjacent to nanolites with a lower ρ_e represent the residual melt depleted in Fe and Ti and enriched in Si and Al. Following nanolite crystallisation, the melt will be enriched in Si and Al and will, therefore, have a lower density than the bulk composition, where this change in both melt composition and density is reflected in the variation in ρ_e . We have now revised the presentation of the results to highlight these areas of lower ρ_e in Figure 4 and provide a revised interpretation which focuses on the change in melt composition and density following nanolite crystallisation (lines 217-228). We have also updated the paragraph which discusses the ρ_e data in the later section of the discussion (lines 290-291). Lange (1997) has been added to the references section.

The reason this is important is that if the nanolites formed in situ, i.e. from the melt with which they are currently associated, this is the pattern one would expect. If they have migrated through the melt they may have been able to shed their FeTi-depleted halo, which one would expect to lead to different properties of the bulk magma.

We have added a more detailed interpretation of the results and the in situ crystallisation of nanolites from the melt to the discussion at lines 424-426.

e.g. On lines 263-270 it is again suggested that “the glass in the nanolite-bearing samples is likely of higher ρ and a different composition” but the example from the literature used to justify this is one where the nanoparticles are surrounded by a “Si-rich ‘shell’ ($\rho = 2.2$ g/cm³)”. In that example, the matrix has a density of 2.7 g/cm³, so the Fe-depleted shell is lower density. There is no reason to think that is not also true for the samples presented here, as detailed in the previous paragraphs. For the shell to be higher density than the surrounding glass would require some remarkable geochemical behaviour that would require additional explanation.

We agree with the reviewer and have revised the sentence to highlight the occurrence of glass with a lower ρ_e (observed in Figure 4), which likely corresponds to the SiO₂ and Al₂O₃ enriched residual melt following crystallisation (lines 290-296). This is in agreement with the reference provided, where less dense, Si-rich shells are observed surrounding Fe-Mn-O nanoparticles.

e.g. Line 305. Normal-sized crystals, which are also surrounded by haloes of depleted melt (boundary layers), still manage to grow. So there may be additional justification needed to assert that the halo prevents further growth of nanolites. And does “gas escape” here refer to actual bubbles (nanobubbles?) or to the diffusion of volatiles, which will also be enriched in the halo? This needs clarifying because there are structures in Figure 2 ascribed to the collapse of bubbles that have escaped.

We have now revised the sentence to indicate that the formation of compositional boundary layers with a more evolved composition, depleted in the elements compatible with titanomagnetite, may reduce the rate of crystal growth and limit further crystallisation (lines 332-334). What we would like to suggest is that the change in melt composition adjacent to nanolites may reduce the rate of crystal growth due to the depletion in compatible elements, rather than preventing further growth. We have revised the sentence to clarify this. The reference to 'gas escape' refers to the decoupling of gas bubbles from the melt, as discussed in references Di Genova et al. (2020) and Kennedy et al. (2022) which are cited in this section of the manuscript. The crystallisation of Fe-Ti oxide nanolites has been shown to facilitate the heterogeneous nucleation of bubbles and maintain gas-melt coupling due to the higher bubble number density. However, this sentence has now been revised and the reference to gas-melt decoupling removed as it is not explored further in the text and does not relate directly to our results on nanoscale aggregates.

One important thing to note is that the halo is depleted in elements that formed the crystal, but enriched in the other elements relative to the adjacent melt reservoir. Chemical diffusion along this gradient will then drive homogenization, so one important question is the timescale of quenching – in the lab or in a Plinian eruption this can be quick, preserving the haloes.

We thank the reviewer for this comment and have added a sentence at lines 340-343 to highlight that the small size of the lapilli clasts examined in this study likely experienced rapid quenching following ejection from the vent, on the order of seconds to minutes, preserving the haloes observed in the matrix glass.

e.g. lines 308 to 314. The presence of undetectable ultrananolites is one possibility, another is dissolution of previously formed nanolites (although why this would happen would need explaining). A third possibility is that the haloes are in fact less dense than the surrounding melt – as appears to be the case from Fig. 4 – although that would appear inconsistent with the densities calculated in Table 1.

We agree with the reviewer and have now updated this section of the discussion to better present the results and interpretations of the ρ_e data, also in response to the comments above (lines 217-228; 290-296). Instead, areas with an ρ_e intermediate between that of the nanolites and silicate glass may indicate the presence of ultrananolites with a size that is below the resolution of the technique (lines 338-340).

Lines 413 to 416. The melt compositions in Supplemental Table 3 also require cpx crystallization, but this should be mentioned explicitly earlier on, when reporting the melt compositions and attempts to identify the nanolite minerals. Is it possible that the phase contrast between clinopyroxene nanolites and residual glass would be so small as to be essentially undetectable with this technique? Could this potentially resolve the problem of a dense melt halo (lines 204-212)?

From the BSE images (Figure 2) and results of modelling using Rhyolite-MELTS, the melt should also crystallise clinopyroxene nanolites. The measurement of the nanolite crystal fraction from the BSE images is 0.27, which indicates that there may also be nanoscale clinopyroxene, as complete extraction from the melt may only crystallise a few vol.% of Fe-Ti oxide (lines 163-172). This agrees with the melt compositions in Supplementary Table 3 (now 1), as clinopyroxene crystallisation is also needed to produce the more evolved composition of the residual melt. Clinopyroxene nanolites have also been observed in volcanic samples by previous studies (Sharp et al., 1996; Mujin et al., 2017). We have added a sentence earlier in the manuscript which references Supplementary Table 1 and specifies

how the change in melt composition may result from crystallisation of both Fe-Ti oxide and clinopyroxene nanolites (lines 172-173).

However, we do not observe clinopyroxene in the orthoslices from X-ray ptychography, only titanomagnetite. As it is possible that it may be difficult to resolve clinopyroxene in the images due to the phase contrast, we also calculated the expected atomic density for clinopyroxene using the measured atomic number in clinopyroxene from the MTL samples and the density for clinopyroxene (3.2 g/cm^3), to obtain quantitative information on the phase. However, the calculated atomic density of the melt halo surrounding nanolites does not agree with the estimation for clinopyroxene, but is more compatible with a phase of higher density. Therefore, from the images, it is more likely that the dense areas surrounding nanolites represent areas where there are Fe-Ti nanolites, but of a scale which is not resolvable using the technique (lines 338-340 in the revised manuscript).

[3] The discussion of the effects of agglomeration probably needs some refining.

e.g. Line 133 how does agglomeration of existing particles produce “chemical changes in the surrounding melt which markedly increases their overall contribution to magma viscosity”? Nucleation and growth of new particles would cause changes in the surrounding melt due to depletion of Fe and Ti, but is there a reason for agglomeration of existing particles (presumably with their own depleted haloes) to cause further changes?

We agree that it is not the agglomeration of the particles which results in changes in the composition of the surrounding melt, but the crystallisation of the nanolites. We have now revised the sentence (lines 134-138) so that it is clearer that it is the crystallisation of the nanolites that results in changes to the melt composition and aggregation is a separate process, but both processes contribute to their impact on magma viscosity.

e.g. lines 326 to 328. “... agglomeration will increase the effective volume of nanolites. Comparing the segmentation of the nanoscale aggregates both with, and without, the incorporation of the surrounding glass (Fig. 6), the initial volume of the aggregate increases by up to 60 %.”

But Fig 6 is comparing the agglomerate with and without its depleted melt halo. The proper comparison is between the agglomerates and the sum of the individual particles with their individual haloes. Agglomeration could result in a smaller volume of enriched melt, because the agglomerated particles presumably have a smaller total surface area than the sum of the individual particles. If the depleted halo compensates for this by thickening, then perhaps there is simply no change in its effective volume.

The effective volume could increase if additional (undepleted) melt were trapped between the particles due to agglomeration. For example, random close packing of uniform spheres yields ~36% space between the spheres. So perhaps the effective volume could be 36% higher, but if the particles need to touch to stick (as van der Waals forces decay extremely rapidly with increasing distance), much of the interstitial melt is likely the depleted halo material.

In short, if a clearer way can be found to explain how agglomeration leads to an effective volume increase, and what the quantitative magnitude of that increase could be, it would be appreciated.

We agree with the reviewer and have now considerably revised the discussion of aggregation in the manuscript to provide a clear description of the process and evaluation of the different mechanisms which may have produced the nanolite aggregates observed in the

samples. This revised section can now be found at lines 472-519 and includes the new Figure 9. We have added several references which discuss the different mechanisms which can lead to the formation of aggregates, such as growth mechanisms (epitaxy and heterogeneous nucleation) and dynamic processes (synneusis). Following the revision of the section which discusses the nanolites and the depleted melt haloes, we also indicate that the boundary layer surrounding nanolites relates to their crystallisation and is produced from the extraction of Fe and Ti from the melt. Therefore, this interstitial glass relates to the crystallisation of the nanolites and is not entrapped between nanolites as a result of a dynamic process such as synneusis. In the revised discussion we present a clearer explanation of the formation of the boundary layer and that the aggregates observed in the samples most likely result from a growth mechanism. Therefore, as also mentioned by the reviewer, we acknowledge that the representation of the volume increase in the original version of the manuscript is not consistent with the observations from the images. Instead, what we would like to indicate is that incorporation of the interstitial melt (or boundary layer) increases the volume of the aggregate and its impact on the magma viscosity (Di Genova et al., 2020). As the depleted melt halo surrounding nanolites is enriched in Si and Al, it will increase the effective volume, as the rheological behaviour will be closer to an elastic response at eruptive temperatures, due to the stable network of Si and O Bonds (Di Genova et al., 2020b; Kennedy et al., 2022). In the manuscript we have now revised the description of this volume increase to clarify this. The specific lines 326-328 highlighted by the reviewer in the original version of the manuscript have now been moved to the section which discusses the formation of the boundary layer, to better highlight that the segmented boundary layer is related to the crystallisation of the nanolites and is not only entrapped between them (lines 330-332, highlighted in orange).

We have also revised the section of the discussion which presents the impact on magma viscosity to specify how nanolite aggregation may increase magma viscosity, with specific reference to the data provided in this study. Nanolite crystallisation and aggregation increase the aspect ratio of the nanoscale particles in the sample, which can be measured in our 3D images and may lead to the development of non-Newtonian rheological behaviour at higher crystal fractions. We have added further references and discussion of this particular impact on magma viscosity at lines 379-388, including the new Figure 8. We thank the reviewer for this comment as this has resulted in a clearer presentation of the aggregation mechanism and impact on magma rheology in the revised manuscript.

[4] The discussion of calculated viscosities needs careful inclusion of uncertainties and better explanation of the assumptions being made

Lines 348 to 352 – please give the model uncertainties on these calculations. For GRD the RMSE for volatile-bearing melts is ± 0.46 log units (their Table 3) so combining that with the ± 0.17 log unit RMSE for Langhammer et al. (2021) means the calculated melt viscosity is 9-19 Pas (Langhammer, best estimate 13 Pas with ± 0.17 log unit rmse) or 21-176 Pas (Giordano with ± 0.46 log units rmse). Using the ± 0.74 log unit RMSE for the GRD model applied to the Langhammer et al. dataset, gives a range of 11-335 Pas, which is already a substantial overlap with the Langhammer model. Given that RMSE is a one sigma uncertainty, the two models in fact agree quite well, even though that gives a total range of 9-335 Pas (or 0.9-2.5 log units). I do agree that it is useful to show how much uncertainty there still is even for predicting melt viscosity.

We have now added the uncertainties for the calculations of liquid melt viscosity from the model (lines 358-359; lines 362-364 and Supplementary Information). To better present the

results of the viscosity estimation for the MTL and include the uncertainties, accounting also for the effect of nanolite crystallisation, we have also revised this section of the discussion, reducing the number of different models and calculations that are presented in the main text. As the results of the GRD and Langhammer models show good agreement for the MTL composition and the conditions used in the estimation of melt viscosity when considering the uncertainty of both models, we have moved this comparison to the Supplementary Information, to provide further information on the uncertainties associated with these calculations. This revision allows a better presentation of the results of the model and the associated uncertainties in the manuscript.

Line 350 – why assume 2 wt.% H₂O – how was this water content determined (by previous studies)? For the evolved melt composition, are you also using 2 wt% H₂O? Crystallization of 27 vol.% crystals should increase the water content in the melt by ~35%, and this should be factored into the magma viscosity calculations.

The water content used in the calculation represents the maximum H₂O content measured in melt inclusions in scoriae of the MTL eruption using Raman spectroscopy (Bamber et al., 2020) and represents the pre-eruptive H₂O content. This information has been added to the manuscript at lines 355-358.

Regarding the calculation for the evolved melt composition, the H₂O content used is 2 wt.% as we interpret that the nanolites crystallise during ascent within the conduit. We agree with the reviewer that crystallisation will increase the H₂O content of the melt, however, for the calculation of the melt viscosity, using this higher H₂O content assumes that the nanolites crystallise within the magma reservoir before magma ascent. Due to the morphology and aggregation of nanolites and the results of the Rhyolite-MELTS simulations (lines 164-166; lines 422-425), we can suggest that the nanolites crystallise during ascent. During magma ascent H₂O will exsolve as crystallisation occurs, so the melt H₂O content will not be as high as expected from 35 % crystallisation.

Lines 357-361 I think the text is saying that all existing viscosity measurements (and hence models) are suspect because nanolites may have affected them. For the specific point being made, that would likely affect measurements of both the initial liquid and the depleted halo liquid – there is still plenty of Fe and Ti in the halo (Supp Table 3). But it is also worth remembering that crystals will not form below the liquidus, and that fast cooling rates seem to be required to form nanolites rather than microlites – which are easily detected by microscopic inspection. Nanolites form at cooling rates of several °C/s (Di Genova et al. 2020) but viscosity measurements by e.g. micropenetration or parallel-plate viscometry typically use much lower cooling rates. And even if they were to form, a sudden nanolite nucleation event should be easily detected in the viscosity data as a discrete jump, given the strong effect they can have on viscosity. So on balance this argument does not feel well justified.

In the manuscript we refer to studies which observe the formation of nanolites during heating and hold at high temperature during viscosity measurements at ambient pressure (Bondar et al., 2025). Recent studies indicate that during viscosity measurements performed by micropenetration and parallel plate viscometry, restructuring of the melt can occur at the nanoscale at temperatures around T_g in the supercooled liquid state, including the formation of nanolites. As our study focuses on 3D observations of nanolites in volcanic samples using multi-scale imaging techniques, we do not describe the results of these works in detail in the manuscript and would like to only highlight the impact of nanocrystallisation during laboratory measurements and how this may contribute to the uncertainty in estimating melt viscosity. We have now revised part of the discussion (lines 366-369) to instead highlight how these

observations may contribute to the uncertainties associated with the estimates of melt viscosity provided in this study.

Line 363 please provide the actual equation for relative viscosity as a function of crystallinity that was used here. Which of the approaches used in Vona et al. (2011) was used here? These include a modified Krieger-Dougherty equation from Ishibashi (2009), the Costa et al. (2009) model, and then a new parameterization based on crystal aspect ratio and strain rate. All three of these are strain-rate-dependent and so the strain rate used in the calculations needs to be stated.

We used the approach of Vona et al. (2011) and the equation of Costa et al. (2009) to calculate the relative viscosity increase due to crystallisation. In Vona et al. (2011), the parameters derived from fitting the data using the Costa et al. (2009) equation are fit irrespective of the strain rate. We use the same fit parameters as Vona et al. (2011) in our calculations of the relative viscosity increase. We have provided further details on the approach used to calculate the viscosity increase in the manuscript (lines 372-374) and Supplementary information, and have added Costa et al. (2009) to the reference list.

Line 373 To summarize, the calculated evolved melt viscosity is 35 Pas (actually 24-52 Pas allowing for ± 1 rmse from Langhammer) or 320 Pas (actually 110-920 Pas allowing for ± 1 rmse from Giordano). The calculated magma viscosity is 150-1400 Pas (Mader model) or 1400-12000 Pas (Vona model). The estimated increase is then a factor of ~ 4 -5 (Mader) or 40-50 (Vona), but not 10-100. Bearing in mind the effect of increased water content in the evolved melt, which is not considered here, the actual viscosity change is likely to be less than this. It is still obviously an important change, but the way things are written seems to always be pushing the limits of the maximum effect that could be claimed, while ignoring the uncertainties in all of these models.

We have updated the presentation of the evolved liquid viscosity calculation in the main text to include the model uncertainty (lines 362-364). We have now revised and rearranged the section of the discussion which presents the results of the viscosity calculations (lines 354-388), to provide a clearer presentation of the results of the models, calculating first the liquid melt viscosity, then the incorporation of crystal fraction and aspect ratio, using our 3D data. This section is now also supported by a more expanded discussion of these uncertainties, comparison of the different models used to calculate the liquid viscosity and justification of the values selected for the parameters included in the relative viscosity models, which can be found in the Supplementary Information.

The sentence in the original version of the manuscript which summarised the results of the viscosity calculation presented the viscosity increase considering both the change in melt composition resulting from their crystallisation and also their contribution as solid particles. The comparison made was between the initial basaltic composition and the results of the Mader and Vona models, resulting in a relative increase in viscosity of a factor of 20-200. We have now clarified this in the manuscript to specify that both effects are considered (lines 376-378). We have also added references to the Supplementary Information in the revised discussion and stated where the evaluation of the models can be found.

Line 380 there needs to be additional context on the basalt with 30% nanolites reaching a viscosity of 10^6 Pas. What temperature, composition, water content (if any) and what would the crystal-free melt viscosity be at those conditions?

The basalt is a basaltic composition of Etna, produced as an experimental starting material from material of the 2001 eruption. The composition is reported in Polacci et al. (2018). The initial melt viscosity of this composition was estimated as 200 Pa s, at pre-eruptive

conditions (1070 °C; 3 wt.% H₂O). These conditions are comparable for the MTL example in this study, which has similar estimated pre-eruptive conditions of 1080 °C and 2 wt.% H₂O. This information has been added to the manuscript at lines 395-398.

Likewise additional context is needed for stating that 60% microcrystals are needed to achieve the same viscosity as 30% nanolites. A quick look at e.g. Figure 2 of Costa et al. (2009) indicates there is a huge uncertainty in the experimentally determined effect of microcrystallinity on effective magma viscosity. Lejeune and Richet (1995) documented a relative viscosity of more than 1000 at only 40% crystals in one experiment, so at the very least there needs to be acknowledgement of the uncertainties in these statements.

The calculation of the relative viscosity increase due to microcrystals, where 60 % microcrystals are required to achieve the same viscosity increase as 30 % nanolites was calculated considering SiO₂ microspheres, also in silicon oil. The microspheres used in the calculation have a similar shape to the nanospheres, and, therefore, a similar aspect ratio. The calculation was performed using the approach of Mader et al. (2013) and the data of Mueller et al. (2010). However, the relative viscosity increase using microspheres was calculated with a strain rate of 1.0 s⁻¹. In Di Genova et al. (2020), less than 10 % nanospheres are required to reach the same relative viscosity increase at these higher strain rates of 1.0 s⁻¹. We have added further details to the manuscript at lines 398-402 to better present and contextualise the comparison between the impact of nanocrystals and microcrystals on the relative viscosity increase.

We also acknowledge that the calculations of the relative viscosity increase due to crystallisation have uncertainties. Figure 2 of Costa et al. (2009) shows a large variation in the observed viscosity increase when comparing the results of different experimental datasets for the same crystal volume fraction. This variation arises from uncertainties in the experimental datasets used in the model regarding the experimental conditions, such as the strain rate or relative proportion of melt and crystals, leading to a large uncertainty in the calculated effective magma viscosity using the model. We have now added further information which details the uncertainties associated with this calculation to the Supplementary Information and a reference in the main text to where this information can be found at lines 352-353.

Line 384 relative viscosity increase is mentioned here but the starting viscosity is not – what was the relative viscosity increase documented by Di Genova et al., and what actual viscosity would that imply starting with the melt compositions and viscosities discussed above?

In Di Genova et al. (2020) the initial viscosity at pre-eruptive conditions for the Etna composition is 200 Pa s. Incorporating the viscosity increase due to nanolites as calculated from the results of the analogue experiments, the viscosity increases to 10⁶ Pa s, depending on the particle content and the strain rate. At strain rates between 1 and 3.5 s⁻¹, this viscosity is reached with a particle volume of only 10 %, whilst at lower strain rates of 100 s⁻¹ it is reached at a particle volume of 30 %. The calculated liquid viscosity of the MTL composition ranges between 13 and 61 Pa s depending on the model used. As the MTL composition and pre-eruptive conditions are comparable to Etna, we may consider that a similar increase in viscosity may occur when considering both the nanolite content and the effect of their crystallisation, based on the results of the analogue experiments and recent studies on volcanic melts (Valdivia et al., 2025). A high strain rate during ascent of 1 s⁻¹ is estimated for highly explosive basaltic Plinian eruptions, so we can assume that this viscosity increase is reached for the MTL case even at lower nanolite crystal fractions than the 0.27 estimated from the BSE images. We have added further information to the revised manuscript (lines

395-405) to better present the comparison with the experimental results of Di Genova et al. (2020) and the implications for the MTL composition.

Lines 393-397 “The micro-textures observed within the lapilli may, therefore, represent features which developed prior to magma fragmentation”.

It is tempting to assume that the crystal textures all formed pre-fragmentation, but wouldn't these clasts have cooled (post-fragmentation) at almost exactly the rates used to stimulate nanolite crystallization in previous experiments such as by Di Genova et al. (2020)? An estimate of the cooling rate could be made based on the pyroclast thermal model of Moitra et al. (2018 G-cubed). For 2-3 cm clasts the cooling rate should be $\sim 4^\circ\text{C/s}$ from 1000 to 600°C . Processes in the conduit prior to fragmentation would likely involve much slower cooling, at ranges typically associated with microlite growth. This should perhaps be stated more clearly.

Although we agree with the reviewer that rapid cooling of clasts post-fragmentation can occur, with the potential to crystallise nanolites, nanolite crystallisation has also been shown to occur over fast timescales, at higher cooling rates than those estimated by Moitra et al. (2018) for cooling of clasts in the atmosphere. The in situ SAXS-WAXS experiments of Di Genova et al. (2020) illustrate that nanolites crystallise rapidly at near-liquidus temperatures, as nanolites formed at temperatures above 890°C for the Etna basalt composition and not at temperatures close to T_g . Nanolites crystallised at high cooling rates of $10\text{-}20^\circ\text{C/s}$, which is higher than that expected for cooling of clasts ejected from the conduit. Experiments show that nanolites crystallise under conditions of high undercooling, estimated between ΔT 40 and 250°C . These conditions are expected to occur for basaltic Plinian eruptions, where rapid magma ascent, between 5 and 50 m/s, can drive rapid syn-eruptive crystallisation within the conduit under conditions of high undercooling (Arzilli et al., 2019; Di Genova et al., 2020). Rapid ascent rates ($>10\text{ m/s}$) and high undercooling are also expected to occur for the Plinian MTL eruption, driving rapid crystallisation within the conduit (Bamber et al., 2020; 2022). Furthermore, several studies on nanolite crystallisation indicate that nanolite formation most likely occurs in the conduit, based on the results of experiments and textural observations of natural samples (Mujin and Nakamura, 2014; Mujin et al., 2017; Mujin and Nakamura, 2020; Knafelc et al., 2022; Okumura et al., 2022). Therefore, it is possible that nanolite crystallisation occurred during magma ascent. For the MTL eruption, we can assume that it is more likely that these textural features represent syn-eruptive, rather than post-fragmentation processes. We thank the reviewer for this comment as we have now revised the manuscript to better clarify our interpretation of the observed textures as syn-eruptive (lines 414-426; lines 428-431). However, we recognise that it would be beneficial for future studies to examine the crystallisation behaviour of nanolites during and post-fragmentation.

Alternatively, if the timescale of “a few minutes” applies to both nanolite growth and ascent to the surface this implies that magma mixing caused an immediate eruption (with a delay of less than a few minutes). How could these two magmas mix that quickly? Do they have different compositions? Were they stored at different temperatures? If the answer to either or both of these is yes, the rheological calculations should compare the two magmas. Such rapid eruption triggering would imply a sudden degassing event, but what exactly about the mixing process would have caused it?

Scoriae of the Las Sierras-Masaya Plinian eruptions show considerable textural heterogeneity, with a crystal-poor and crystal-rich endmember and enclaves which may indicate the development of a crystal-rich mush at depth within the storage reservoir (Bamber et al., 2020; 2022). The textural heterogeneity observed with the crystal-poor and

crystal-rich endmembers has been interpreted as resulting from possible zonation in ascent velocity within the conduit or syn-eruptive mingling. Therefore, the mingling textures observed may represent syn-eruptive processes instead of magma mixing at depth. As nanolite crystallisation likely occurs within the conduit during magma ascent, we do not discuss possible triggering mechanisms for the eruption in this paper. We thank the reviewer for this comment as it is possible that pre-eruptive conditions have an influence on nanolite crystallisation, however, this discussion would require further data than is provided in this work.

Lines 423-425. The assertion that the MTL magma viscosity did reach 10^6 Pas in the conduit, from Bamber et al. (2020), appears based on a microlite fraction of 0.5 and a starting melt viscosity of 10 Pas (section 4.3.2) or 30-40 Pas (section 4.2.3), and the Costa et al. (2009) model. Costa et al. (2009) present a framework for fitting experimental data but not a specific set of recommended parameters. Mader et al. (2013) suggested a set of parameters for use with Costa et al. (2009) but they are somewhat arbitrary – the data in Fig 2 of Costa et al. (2009) show that there is no single equation (or parameters) that can describe relative viscosity as a function of crystal fraction. The MTL magma viscosity quite possibly could have reached 10^6 Pas, but again there are uncertainties that need to be acknowledged.

We acknowledge that the results of Costa et al. (2009) show that at the maximum crystal fraction measured in the samples (0.5), there can be an order of magnitude variation in the relative viscosity increase estimated by varying Φ^ and γ , indicating that the selection of the parameters can influence the viscosity calculated. For calculation of the MTL viscosity, the values determined by the experiments of Vona et al. (2011) are used as these values were determined from experiments performed on a similar basaltic composition, to assess the impact of crystallisation on magma rheology. We have now added a discussion of the uncertainty associated with the selection of these parameters in the Supplementary Information.*

[5] Miscellaneous minor points:

Line 1 The title might be more accurately written as “3D visualisation of nanolite aggregates in basaltic magmas...” because the process of aggragation is not seen in a static image.

We agree with the reviewer and have now updated the title of the manuscript.

Line 64 “high viscosity facilitates the accumulation of stress due to applied strain”. This may need rephrasing – in volcanic systems the stress is applied and the strain results from that, not vice-versa. High viscosity means a low strain rate and hence low rate of stress dissipation, but it does not cause the high stress in the first place.

We have now revised the sentence to instead indicate that high magma viscosity can enable brittle fragmentation, where fragmentation arises from stress accumulation at high strain rates and bubble overpressure (lines 64-66).

Line 74 how does “fracturing ... modify the rheology of basaltic magma”? Isn't fracturing an indication that the rheology has already changed from the ductile to brittle regime, through some other mechanism (e.g. high strain rate, change in effective viscosity, etc)?

We agree with the comment of the reviewer and have now revised the sentence (lines 74-76). In the previous version of the sentence, the phrase ‘modification of magma rheology’ was in reference to syn-eruptive crystallisation. However, we acknowledge that by including also fracturing in the same sentence, this phrase was in the wrong place. Instead the

connection between syn-eruptive crystallisation and magma rheology is discussed in greater detail in the following paragraph (lines 77-81; highlighted in orange).

Lines 90-92 is it necessary to list “increasing melt polymerization” separately from “chemical and structural changes in the residual melt”? It may be better worded as “by facilitating heterogeneous bubble nucleation or inducing chemical and structural changes in the residual melt, for example increasing melt polymerisation, ...”

We thank the reviewer for the suggestion and have revised the sentence (lines 91-94).

Lines 176-178 I agree that there does not appear to be Si enrichment around the nanolites – and for Fig 3m there does not appear to be much Si depletion at the site of the nanolites either, which is surprising.

In Fig 3m, some depletion in Si can be observed in the areas where the Fe-Ti nanolites occur, relative to the surrounding distribution of Si in the glass. However, this is not as clearly observed as in Fig 3n where there is a more well defined area of Si depletion where the Fe-Ti nanolite aggregates occur. It is possible that it may be more difficult to resolve the differences in Si at this scale, compared to Fe where there is a large difference in the Fe content of the nanolite compared to the glass.

Lines 223-225 and caption to figure 6. The caption should also explain why this part of the glass is shown and how it is distinguished from the rest of the glass (lower density). Where we can see the nanolites in Fig 6 c and d, does this imply a lack of the evolved boundary layer in these areas, or has it simply been cut away to reveal the nanolite inside? This is important to know. The 2D orthoslices suggest that the enriched boundary layer is not present everywhere around the nanolites.

The part of the glass that is shown in the volume renderings in Figure 6 represents the boundary layer depleted in Fe. We quantified a difference in phase shift between the glassy boundary layer around the nano-aggregates and the glass not affected by nanocrystallisation, which is reported below:

Phase shift range for the glass not affected by nanocrystallisation: -0.0046 to -0.0079;

Phase shift range for the boundary layer: -0.010 to -0.013.

This difference in phase shift allowed segmentation and separation of the boundary layer in 3D. Regarding Figure 6c, where some small nanolites are visualised without the boundary layer around them, this is due to the 3D visualisation, as all the nanolites have a boundary layer around them. We have provided a new volume rendering (Supplementary Figure 4) that allows us to visualise the boundary layer around all of the nanolites and updated the caption for Figure 6 (lines 943-945; lines 946-948).

Line 233 why is the unit of ρ_e cm⁻³ and not m⁻³? cgs units should be replaced with SI units wherever possible.

The unit for ρ_e has now been updated to m⁻³ (line 256) and units replaced with SI units throughout the manuscript. Figures 4 and 5 and Table 1 have also been updated to use SI units.

Line 242 does ρ_a have a name (e.g. is it the atomic density), and are the units m⁻³?

ρ_a is the atomic density. The units have been replaced as m⁻³ in response to the comment above. As this information was present only in the methods section in the previous version of the manuscript we have now added it to the main text at line 265.

Lines 297-301 Si⁴⁺ is also a cation, so the boundary layers are not “depleted in cations” but they are often depleted in Fe and Mg or Ti. The key is that the boundary layer is enriched in network-forming Si⁴⁺ and Al³⁺ (don't forget the alumina!). The long sentence on lines 297-301 mentions viscosity increase in two different places, and should be restructured.

We have now revised this sentence and restructured it, so it more clearly focuses on the connection between the chemical changes in the melt and the resulting local increase in melt viscosity. We have also specified that it is the depletion of Fe, Mg and Ti and the enrichment of Si and Al which can lead to this increase (lines 323-327).

Lines 301-304. If you observe shells are 10-350nm in size then that is the size they can reach (at least!). Remove “can reach 10-100 nm in size” on line 301.

We have removed the phrase at line 301 (now lines 328-330).

Line 341 “Nanolites can increase magma viscosity by up to two orders of magnitude”. So can larger crystals - it all depends on the volume fraction, which should be specified here. Unless this is meant to imply an additional increase relative to microcrystals at the same volume fraction, in which case that should be specified clearly.

We thank the reviewer for this comment and have now revised the sentence to highlight that nanolites can increase magma viscosity by up to two orders of magnitude even at low particle contents of < 5 vol.% (lines 392-394). We agree that it is important to state the volume fraction here as the impact of nanolite crystallisation on magma viscosity is considerable at much lower crystal contents relative to microlites. This is due to their small size, which influences liquid flow between particles and has a larger impact on magma viscosity compared to microlites at the same crystal content. Nanolite crystallisation may also lead to an increase in the melt viscosity due to the effect of their crystallisation on the melt chemistry and the formation of nanoscale chemical heterogeneities (Valdivia et al., 2025). This comparison with microlites and the impact on viscosity at the same crystal content is further discussed at lines 395-402.

Line 480 Should the beam diameter be 10 nm rather than 10 microns for the BSE imaging? A 10 micron diameter would be good for a glass microprobe analysis but not for high resolution imaging.

Yes, the beam diameter during the image acquisition is ~10 nm. The conditions reported in the manuscript refer to the glass analysis performed on the same area as the image. As this is not representative of the conditions during image acquisition, the beam diameter is not specified in the revised manuscript (lines 561-562).

Lines 561-562 should the average atomic numbers be given as 12.91±0.07 and 20.32 ± 0.04? Please specify whether the ± is one standard deviation or two.

We have revised the presentation of the atomic numbers in the Methods section at lines 642-643.

Fig. 2 is described in the caption as showing two pieces of scoria from the FL eruption, and the image labels are “FonAB” and “FonE”. These appear to be two different layers of the FL eruption but this should be clarified. Moving the sample descriptions from supplemental material into the main text would help with this.

We agree with the reviewer and have now moved the sample descriptions from the Supplementary Information to the Methods section of the main text, in the section ‘Sampling’ (lines 544-557), to provide further context for the samples used in the study. We have also

added a line to the Methods section to explain the nomenclature used for the samples and how these relate to the Fontana Lapilli eruption sequence and stratigraphic units (lines 556-557).

Fig. 5 it would be helpful to see MTG at the same spatial scale as MTL-A is shown in Fig. 4 (as they were collected using the same geometry and resolution). To help make the comparison described in the text, this could become a third panel of Fig. 4, and the greyscale should be the same in all three panels (currently the greyscale range in Fig 5 maps to a different range of electron densities compared that used in Fig 4). Please also explain the cause of the “thumbprint” pattern evident in Figure 5 – is this some kind of artifact arising from the ptychographic technique or the image processing?

The greyscale range is different for Figure 5 (MTL-G) as using the same grayscale range as in the other samples results in saturation of the lowest electron density values which correspond to the air. It is difficult to maintain the same range for the colourbar as in Figure 4 as there is a different range in electron density values across the samples (due to the presence of the nanolites in MTL-A and MTL-B with high values of electron density). However, we agree with the reviewer that it can help to have a panel with comparable scale to MTL-A for comparison of the electron density distribution and heterogeneity in the glass. To help with this comparison, we have now added a new Supplementary Figure (Supplementary Figure 1) which shows sample MTL-G at a comparable scale and more clearly presents that it is a homogeneous glass also at the same scale as the images of MTL-A. This is referenced in the text at lines 218-221.

The ‘thumbprint’ pattern observed in Figure 5 is a grid artefact, which is a known artefact in ptychography that comes from the regular scanning grid. This artefact can then affect the algorithm for producing the 3D reconstructions.

For the supplemental information, it would be helpful for the reader to have the sample information in the main article text.

We have now moved the sample information to the Methods section of the main text, in the section ‘Sampling’ (lines 544-557). The reference Costantini et al. (2009) has also been moved from the Supplementary References to the References in the main text.

Supplementary Tables 3 and 4 would also be useful in the main text, as the glass compositions and calculated viscosities are discussed extensively.

As we have now considerably revised the discussion, adding 2 figures and revising the presentation of the viscosity calculations, we would prefer to still keep Supplementary Table 4 in the Supplementary Information. Although we agree with the reviewer that this table can be helpful in the main text, we believe that Figures 8-9 contribute more to the discussion, as part of the comparison of the viscosity models has now moved to the Supplementary Information. As Supplementary Table 3 summarises chemical data which has already been published and is used only as the input liquid composition for the calculations, we have chosen to keep this table in the Supplementary Information as these data are not new results from this study.

Reviewer #3 (Remarks to the Author):

Review of manuscript NCOMMS-24-72256

Titled: " 3D visualisation of nanolite aggregation in basaltic magmas using X-ray ptychography: Implications for magma rheology"

Authors: Emily C. Bamber, Fabio Arzilli, Silvia Cipiccia, Darren J. Batey, Giuseppe La Spina, Margherita Polacci, Ali Gholinia, Heath Bagshaw, Danilo Di Genova, Richard Brooker, Daniele Giordano, Pedro Valdivia, and Mike R. Burton

Dear Dr. Neely,

Thank you for the opportunity to review this article. The manuscript presents a new approach to volcanic material characterization to provide a careful assessment of the Nanolite content using two explosive basaltic eruptions from the eruption of Las Sierras-Masaya volcanic system as case studies.

Measuring and characterizing Nanolite size and geometry is something that traditional volcanological studies have not done and this technique is a promising step toward understanding more about whether and how important these particles are in terms of determining eruption intensity and style. The presented multipronged approach, using a wide range of complementary analytical techniques is exciting and may help advance our understanding of these phenomena. The presented data and analysis are sound and support the main arguments.

In the annotated pdfs of the main manuscript and supplements I have attached to this letter I provide a range of comments that include minor suggestions for re-structuring and re-wording and some more in-depth analyses / questions about what could be discussed on the basis of these data. I realize that some of those may go beyond what the authors intend to present in this paper but wanted to include them in case there was capacity to address them here or in future studies. I also provide a few suggestions for where and how this work could be improved and where more detail / better reasoning may help. I hope this can serve to further improve this piece of research. Overall, I suggest minor revisions to address relevant comments before publication.

Best Regards,

Stephan Kolzenburg

Line 38: What is the absolute volume increase – and the effect of that on rheology? Nowhere near as drastic as analogue materials suggest I suppose (i.e. a 60% increase to 2% crystals still only brings it to 3.2% total solids) – please document and compare.

The interstitial melt represents the chemically differentiated boundary layer which forms following nanolite crystallisation and is then trapped during the growth of the aggregate. We acknowledge that this increase in effective volume of the aggregate is a localised effect. We have now modified the abstract to specify that this is the differentiated boundary layer surrounding nanolites and that this is a localised effect in the glass, as this is a better representation of the results. We have removed the reference to the effective volume increase within the abstract and conclusion, also in response to a comment from the other reviewer as the explanation of this increase requires further context which is not possible to

provide within the shortened abstract and conclusion. In the manuscript we have now provided a further explanation of this effective volume increase and what this represents at lines 323-332 and lines 504-512. What we can suggest from our results is that the increase in volume of the aggregate due to the differentiated boundary layer may increase their impact on magma rheology, by increasing the effective volume and the local viscosity of the melt. It is difficult to calculate the absolute volume increase attributed to the aggregate considering the total sample due to their localised effect. However, in the revised manuscript we have provided a clearer explanation of the process and better contextualised the presentation of the effective volume increase by moving it to the section of the discussion which presents the boundary layer.

Lines 40-41: How was that determined? Since it was not measured, maybe reword to. "is estimated to have increased..."

We agree with the reviewer and have revised the phrase in the abstract (lines 39-40).

Line 64: Should be called strain RATE. Since there is a time dependence in the relaxation.

We have updated the sentence to specify that this is the strain rate (line 65).

Line 68: Why is SO₂ relevant for eruption style – unclear. Erupted volume or volumetric rate would be more meaningful here.

We have revised the sentence to instead refer to parameters which better relate to the eruptive style (lines 66-69). As there are no estimations of the erupted volume for this event in the literature, due to the complexity of the eruption (Bani et al., 2025), we have chosen instead to refer to the column height. This recent reference has also been added to the reference list.

Lines 77-81: Repetitive – could be simplified

We agree with the reviewer and have now revised this section, removing repetitions (lines 77-81).

Lines 82-83: This quite certainly does not represent the magma during ascent and fragmentation – should be highlighted that these values must be lower in syn-eruptive scenarios

We agree with the reviewer and have revised the sentence to highlight that the high measured crystallinity may not only represent syn-eruptive crystallisation. We have also presented the range in measured crystallinity in the samples rather than just the maximum value, to show the wider range in estimates for crystal content in these samples (lines 82-83).

Line 86: Strikethrough text

We have corrected the sentence (line 87).

Line 87: Strikethrough text

We have corrected the sentence (line 88).

Lines 119-120: Which one? It is implied that it is scale but should be stated explicitly

The main limitation of standard CDI is that the field of view is limited by the lateral coherence length of the beam. Instead, by scanning, ptychography can overcome this limitation,

allowing imaging of extended objects. We have added this more detailed explanation at lines 120-123.

Lines 136-137: No. rheology or behaviour were not measured here. But this work does improve the understanding of nano structure of volcanic products. Please reword accordingly.

We agree with the reviewer and have now revised the sentence to focus on the contribution of our results to the understanding of both the nanoscale structure of volcanic products and the dynamics of basaltic Plinian eruptions (lines 140-143).

Line 163: Unclear if this refers to the crystals or the clusters – please reword for clarity

We have revised the sentence to specify that it is the clusters which form the more irregular shapes (lines 177-178).

Lines 167-168: What causes these differences? Why are some areas showing nanolites and others not – would be interesting to hear some thoughts on that.

Samples of the MTL and FL eruptions show considerable textural heterogeneity, which is observed at the micron-scale where crystallinity varies between 5 and 63 vol.% (Costantini et al., 2010; Bamber et al., 2022). This heterogeneity has been interpreted as resulting from syn-eruptive processes such as gradients in ascent velocity within the conduit during magma ascent or mingling of two endmembers with different thermal histories (Bamber et al., 2020). It is possible that these processes may also affect the nanoscale structure of the magma and its potential to crystallise nanolites, leading to the heterogeneity observed in the samples. We observe this heterogeneity at a larger scale in our samples in the BSE images (Figure 2), which show spatial variations in the size and distribution of nanolites. The distribution of nanolites has also been observed to be associated with bubbles, as shown in the experiments of Pleše et al. (2019), where nanolites nucleate on the surface of bubbles during vesiculation (observed in Fig. 2e-f).

Petrological analysis of basaltic samples also finds that unmixing of magmas into Fe- and Si-rich liquids can also lead to compositional variations and the heterogeneous development of textures at the nanoscale (Honour et al., 2019). In these cases, the formation of compositional boundary layers during crystal growth can produce spatial variations in melt composition, triggering unmixing and/or the development of Fe-rich filaments (Honour et al., 2019; D’Oriano et al., 2025). This may lead to heterogeneous crystallisation of nanolites within the matrix. However, the thermodynamic conditions and kinetics of these processes described above are uncertain at the nanoscale, as there are less experimental constraints. Therefore, as our work focuses on imaging of the morphology and distribution of nanolites within natural volcanic samples, we choose not to focus on the processes which may generate heterogeneities in the ability of the melt to crystallise nanolites without additional constraints. We thank the reviewer for this comment and agree that this is an interesting aspect of nanocrystallisation which could be investigated in further detail in future work.

Line 170: Strikethrough text

We have corrected the sentence (line 185).

Lines 209-211: Evolved in what way? This should be clarified. With Fe and Ti in the oxide nanolites, what is the expected composition of the halo melt. Seeing now that this is a section coming later – maybe refer to that here just to show it will be picked up.

We have revised the sentence to specify what melt chemical changes occur following nanolite crystallisation (lines 222-226).

Line 217: How do 2D volume estimates match the reconstructed 3D volumes? This is a real dataset to shed more light on the 2D-3D comparison at scales that it was never done before – would be good added value to the paper with little to no effort – since the data seem to already exist

We thank the reviewer for this comment and agree that it is valuable to investigate if quantitative textural measurements performed using 2D and 3D images are comparable at the nanoscale. From our 3D ptychographic images, we observe that below a size of ~ 200 nm, nanolites have a low aspect ratio and are approximately spherical-sub-spherical in shape. Therefore, for nanolites of this size, 2D images, such as those obtained with TEM or STEM imaging, should provide a good representation of the morphology, also in 3D. Estimates of the size and volume comparing 2D and 3D images may in this case be similar. Instead, for nanolites larger than this size, the aspect ratio increases and nanolites begin to have a more complex, irregular shape in 3D. This complex morphology may not be captured in 2D images and the estimate of the size and volume of the nanolites may, therefore, differ comparing 2D and 3D techniques.

Comparing the results of quantitative textural analysis using two-dimensional (2D) and three-dimensional (3D) characterization methods, the volume estimates of the nanolites derived from 2D data can be underestimated by one to two orders of magnitude. This discrepancy arises primarily from the limitations inherent in 2D imaging, which may not fully capture the true volumetric size and morphology of nanolites. As discussed above, the extent of the underestimation depends on the nanolite shape; it is particularly pronounced in cases where nanolites exhibit complex, dendritic morphologies formed through aggregation-driven growth. In such instances, the likelihood of a 2D cross-section being representative of the full 3D morphology and size is significantly reduced, leading to misrepresentation of the volume and difficulty in applying an appropriate stereological correction to the 2D data. We have added details on this comparison of the 2D and 3D characterisation methods to the Supplementary Information (Supplementary Table 4) as this provides a valuable comparison of different quantitative textural techniques at the nanoscale. This comparison has been added to the main text at lines 242-249.

Line 230: Strikethrough text

We have corrected the sentence (line 210).

Lines 266-267: State why – to make it very clear to people new to this topic

We thank the reviewer for this comment and have added an explanation at lines 291-293 which specifies that the change in melt composition and, therefore, density occurs in the residual melt due to the depletion of elements compatible with titanomagnetite crystallisation. This is now discussed in further detail in the revised manuscript, also in response to a comment from the other reviewer at lines 217-228 (highlighted in blue).

Line 278: Do you really need all these references to support this simple and well known point

We have reduced the number of references here (line 304).

Lines 302-303: It would be valuable to estimate HOW MUCH this changes the viscosity in the boundary layer to place the later comments on volatile and transition metal diffusion on more solid footing

We agree with the reviewer and considering the comments of both reviewers, we have now made significant changes to the discussion of the viscosity calculations in the revised manuscript. In the discussion we calculate the viscosity of the evolved melt composition and evaluate the impact of this on the magma viscosity. The results of these calculations and the analysis of how the formation of the aggregates can increase the viscosity, considering the change in chemical composition of the boundary layer and the increase in the aspect ratio of the nanoparticles, are provided in lines 346-378 of the revised manuscript.

Lines 306-307: WHY? These higher viscosity zones should have longer relaxation timescales and thus fragment easier at the same change in strain rate than the surrounding melt – please elaborate.

In our work we discuss the results of the experiments of Hornby et al. (2024) on how the presence of nanoscale crystals in the matrix and the consequent development of compositional boundary layers can influence fracture propagation. In that study, the presence of nanoscale crystals is suggested to increase the fracture toughness of the matrix, as fractures branch around nanolites. Instead, fractures propagate preferentially through the adjacent Mg and Fe depleted compositional boundary layers. In our work, what we intend to highlight in this section of the discussion, is that the development of Fe depleted boundary layers following nanolite crystallisation may lead to variations in the fracture resistance of the matrix during fragmentation. However, we agree with the reviewer that these heterogeneous zones within the matrix may be more susceptible to fragmentation under the high strain rates expected during magma ascent for a Plinian eruption. The results of Hornby et al. (2024) are more applicable to cases where fragmentation results from unloading or from secondary processes such as particle collisions, and are more appropriate for less explosive activity, such as Vulcanian eruptions. Although in our manuscript we only intend to highlight how variations in melt chemistry may lead to nanoscale heterogeneity in properties of the glass matrix, such as fracture toughness, we recognise that the example cited may be limited in its applicability to the Plinian case. We have decided to remove the sentence relating to fracture propagation in the revised version of our manuscript (lines 332-334), as this concept is also not discussed further.

Line 328: Please propagate this to estimate the effective volume of these agglomerates (particles + immobilized melt_high viscosity halo)

The calculation of the effective volume of the agglomerate included both the particles and the higher viscosity boundary layer surrounding them. However, in the original version of the manuscript the explanation was not clear. We have now revised this sentence to specify that the glass surrounding the nanolites used in the calculation is the boundary layer (lines 330-332).

Lines 353-361: So all this still places you well below where any ‘traditional’ explosive magma is thought to fragment in terms of viscosity space – how would you generate the strain rates needed to exceed melt relaxation at such low viscosities? Please elaborate.

Here, at this point of the discussion, the viscosity is calculated only for the liquid, without considering the effect of crystals. To calculate the overall rheological impact of the nanoparticle bearing suspension, we calculate both the viscosity of the liquid composition and the relative viscosity increase from the presence of the nanocrystals. The contribution from the crystals is calculated later in the discussion. We consider that fragmentation occurs throughout the magma and not only the melt component and to evaluate the role of the viscosity increase arising from nanocrystallisation in magma fragmentation and eruption dynamics we calculate the mixture viscosity (melt + crystals).

Accounting also for the presence of crystals (nano- and micron-scale), the magma viscosity is 10^6 Pa s, which reaches the fragmentation threshold, considering the results of experiments and numerical modelling on basaltic magma compositions (Papale, 1999; Arzilli et al., 2019). Images of fractures and analysis of their distribution in natural samples from explosive basaltic activity indicates that brittle failure of basaltic magma can occur through the propagation of tensional fractures throughout the magma (Taddeucci et al., 2021). Therefore, for our MTL composition, we can also consider that brittle fragmentation occurred throughout the magma and account for the role of the mixture viscosity in increasing the potential of magma fragmentation. The calculations of liquid viscosity and the relative viscosity increase arising from the presence of crystals was present in the original version of the manuscript. However, we have now considerably revised and restructured the presentation of the viscosity calculations in the discussion of the revised manuscript (lines 346-388) to better present our results. We have also added an introductory section to this part of the discussion to highlight that our calculations consider both the liquid viscosity and the contribution from crystals and it is the mixture viscosity which overall contributes to the potential of magma fragmentation during ascent (lines 346-352).

Line 365: Nearly 30 vol.% nanolites from 13 wt.% Fe in the melt – somehow that does not seem to add up. I think it would be worth to calculate the max volume fraction of Fe-Ti oxides that the melt can produce (assuming all Fe and Ti goes in to these and then see if/how these numbers compare with the 27% reported here).

According to the results of simulations using Rhyolite-MELTS, the maximum volume fraction of Fe-Ti oxides which can crystallise from the composition is 0.15. We agree with the reviewer that the calculation of 0.27 is not compatible with only the crystallisation of Fe-Ti oxide nanolites. In our estimate of the nanolite crystal fraction of 0.27, we consider also the crystallisation of clinopyroxene nanolites, which is evidenced from the melt composition in Supplementary Table 1 and the crystallising phases predicted by Rhyolite-MELTS calculations. However, the discussion of clinopyroxene nanolite crystallisation occurred in a later part of the manuscript, after this calculation in the original version. We have now updated the presentation of these results in the revised manuscript so they occur in the results section and earlier parts of the discussion (lines 163-169), as this is an important result for the explanation of the measured crystal fraction. We have added the estimate of the maximum volume fraction of Fe-Ti oxide which can crystallise from this melt at lines 169-172.

Line 371: This is not Mader's model. I think it would be better to say using the approach presented in Mader and then cite the model you end up using (which were reviewed and placed in concert in the Mader paper).

We agree with the reviewer and have now updated the manuscript and Supplementary Information to specify that the Krieger and Dougherty (1959) equation was used to calculate the relative viscosity, following the approach of Mader et al. (2013). Following the revision of the discussion, this calculation now occurs in the Supplementary Information. We have added the Krieger and Dougherty (1959) reference to the supplementary references list.

Lines 413-414: How do you merge this with the reported nanolite content of 27%? Should be commented on.

The results of the simulations suggest that the measured crystal content of 27 % likely consists of both clinopyroxene and titanomagnetite. This is likely as complete extraction of Fe and Ti from the melt may still only crystallise a small vol.% of titanomagnetite. We have

updated the manuscript to compare the results of the simulations and textural observations and the implications for the mineralogical assemblage (lines 163-172).

Line 424: Insert 'may have'.

We have updated the sentence (line 443).

Lines 424-425: Where does the 10^6 come from now? Unclear-please expand on this for clarity. Above you only say that in other systems this was suggested but the data presented here provide no foundation for this statement other than a vague comparison – needs to be substantiated or significantly toned down.

The estimate for effective magma viscosity here is from a previous study on the MTL eruption (Bamber et al., 2020), which instead analysed micron-scale crystals and the impact on magma viscosity. We have revised the sentence (lines 443-446) to better clarify that this is an estimate of magma viscosity from a previous study, which quantifies the impact of the micron-scale fraction on magma viscosity, and also that microlite crystallisation may have increased the potential of magma fragmentation, as we recognise that there are other syn-eruptive processes during ascent which can influence eruption dynamics. The samples examined in this study and Bamber et al. (2020) are from the same deposit and same unit of the MTL eruption, however, here we examine the microlite-poor regions of scoriae where the glass contains nanolites.

Line 437: But a paragraph before this is estimated at 10^6 – which is it now?

Here, the estimate for the magma viscosity refers to the microlite-poor regions of the MTL samples. The MTL scoriae exhibit significant textural heterogeneity in terms of crystallinity, where the microlite crystal fraction varies from 0.2 to 0.5. The estimated viscosity of 10^6 refers to the microlite-rich regions of the MTL samples. We have now revised this section (lines 455-460) to better differentiate between the different estimates of magma viscosity for the MTL, accounting for the heterogeneity observed in the samples.

Line 781: There are authors listed that seemingly had no contribution to the paper – at least following this statement. Either add their contribution or remove them from the author list.

Further details have been added to the author contributions statement.

Line 821: Add numbers to scale bars in figure

We have updated the scale bar for Figure 4.

Line 826: Add numbers to scale bars in figure

We have updated the scale bar for Figure 5.

Supplementary Information

Supplementary Table 3: Where did these elements go? What other crystals formed?

It is possible that crystallisation of clinopyroxene nanolites also occurred, as shown by the results of the quantitative textural measurements on the 2D BSE images and Rhyolite-MELTS modelling, indicating that the measured crystal fraction of 0.27 likely does not represent only titanomagnetite. Complete extraction of Fe from the melt will only crystallise a few vol.% of titanomagnetite (lines 169-172). We have now added a sentence to the revised manuscript (lines 172-173) which clarifies that the observed change in melt chemistry may

result from the crystallisation of both titanomagnetite and clinopyroxene nanolites, with specific reference to Supplementary Table 1 (highlighted in blue).